# Anthropogenic climate change drives non-stationary phytoplankton internal variability

Geneviève W. Elsworth[1], Nicole S. Lovenduski[2], Kristen M. Krumhardt[3], Thomas M. Marchitto[1], and Sarah Schlunegger[4]

[1]Department of Geological Sciences and Institute of Arctic and Alpine Research, University of Colorado Boulder, Boulder, Colorado, USA
[2]Department of Atmospheric and Oceanic Sciences and Institute of Arctic and Alpine Research, University of Colorado Boulder, Boulder, Colorado, USA
[3]Climate and Global Dynamics Laboratory, National Center for Atmospheric Research, Boulder, Colorado, USA
[4]Department of Atmospheric and Oceanic Sciences, Princeton University, Princeton, New Jersey, USA

**Correspondence:** Geneviève Elsworth (genevieve.elsworth@colorado.edu)

**Abstract.** Earth System Models suggest that anthropogenic climate change will influence marine phytoplankton over the coming century, with light limited regions becoming more productive and nutrient limited regions less productive. Anthropogenic climate change can influence not only the mean state, but also the internal variability around the mean state, yet little is known about how internal variability in marine phytoplankton will change with time. Here, we quantify the influence of anthropogenic climate change on internal variability in marine phytoplankton biomass from 1920 to 2100 using the Community Earth System Model 1 Large Ensemble (CESM1-LE). We find a significant decrease in the internal variability of global phytoplankton carbon biomass under a high emission (RCP8.5) scenario, with heterogeneous regional trends. Decreasing internal variability in biomass is most apparent in the subpolar North Atlantic and North Pacific. In these high-latitude regions, bottom-up controls (e.g., nutrient supply, temperature) influence changes in biomass internal variability. In the biogeochemically critical regions of the Southern Ocean and the Equatorial Pacific, bottom-up controls (e.g., light, nutrients) and top-down controls (e.g., grazer biomass) affect changes in phytoplankton carbon internal variability, respectively. Our results suggest that climate mitigation and adaptation efforts that account for marine phytoplankton changes (e.g., fisheries, marine carbon cycling) should also consider changes in phytoplankton internal variability driven by anthropogenic warming, particularly on regional scales.

## 1 Introduction

Anthropogenic climate change significantly impacts marine ecosystems from phytoplankton (Bopp et al., 2001, 2013; Laufkötter et al., 2015; Kwiatkowski et al., 2020) to fish (Perry et al., 2005; Cheung et al., 2009, 2010; Mills et al., 2013; Wernberg et al., 2016; Flanagan et al., 2018; Staudinger et al., 2019). As the base of the marine food web, phytoplankton support diverse marine ecosystems by providing food for higher trophic levels (Falkowski, 2012). Constraining future changes in phytoplankton with anthropogenic warming is important at regional scales for fisheries adaptation (Pauly and Christensen, 1995; Chassot et al., 2010; Link and Marshak, 2019; Marshak and Link, 2021), particularly as phytoplankton biomass is incorporated into offline fisheries models to predict changing catch potential (Christensen and Walters, 2004; Travers-Trolet et al., 2009; Lehodey

et al., 2010; Maury, 2010; Blanchard et al., 2012; Christensen et al., 2015; Jennings and Collingridge, 2015; Tittensor et al., 2018; Petrik et al., 2019; Heneghan et al., 2021). In this context, understanding changes in both phytoplankton biomass and its internal variability is essential in reducing uncertainty in marine ecosystem projections.

The abundance and distribution of phytoplankton, the base of the marine food web and an important component of the marine carbon cycle, will likely change with anthropogenic warming. Future projections of climate change impacts reveal a global loss of marine net primary production (NPP) and phytoplankton biomass, particularly at middle and low latitudes (Steinacher et al., 2010; Bopp et al., 2013; Lotze et al., 2019; Tittensor et al., 2021). A majority of Earth System Models (ESMs) project an increase in phytoplankton abundance in the high latitude ocean as light limitation is alleviated by stratification, increasing

temperature stimulates photosynthesis, and sea ice cover declines (Steinacher et al., 2010; Bopp et al., 2013). In contrast, a decrease in the low latitude oceans is projected as nutrient limitation from thermal stratification is enhanced (Steinacher et al., 2010; Kwiatkowski et al., 2020). While bottom-up controls (e.g., nutrient flux, light availability) have been shown to affect phytoplankton growth in a changing climate, top-down controls (i.e., zooplankton grazing) also play a role. For example, analysis across a suite of models forced under climate change scenarios reveal grazing pressure as a driver of biomass decline

in low to intermediate latitude regions (Laufkötter et al., 2015). Additionally, top-down controls have been shown to affect regional changes in NPP and export production (Bopp et al., 2001), as well as the timing of phytoplankton bloom onset (Yamaguchi et al., 2022). Regional redistributions of phytoplankton biomass have consequences for fisheries management and conservation (Blanchard et al., 2017; Stock et al., 2017), and may have implications for economics and policy making decisions (Moore et al., 2021).

While climate change is known to impact the mean state of phytoplankton biomass or NPP (Bopp et al., 2013; Kwiatkowski et al., 2020), less is known about how climate change will affect internal variability in these quantities. One recent modeling study found that climate change alters the timing of seasonal blooms in many regions of the global ocean, an effect that could be realized by the end of the century (Yamaguchi et al., 2022). Several other recent studies have demonstrated how other aspects of the coupled atmosphere-ocean climate system are projected to experience changes in internal variability in a changing climate

(Resplandy et al., 2015; Landschützer et al., 2018; Kwiatkowski and Orr, 2018; Rodgers et al., 2021). For example, Resplandy et al. (2015) examined the contribution of internal variability to air-sea $CO_2$ and $O_2$ fluxes with climate change using a suite of six ESMs. Their analyses revealed distinct regional differences in internal variability of air-sea gas fluxes, with the Southern Ocean and the tropical Pacific playing a significant role. Other studies have revealed increases in the frequency of modes of internal variability such as El Niño and La Niña events in response to greenhouse warming (Timmermann et al., 1999;

Cai et al., 2014, 2015, 2022). Clarifying how internal variability in phytoplankton biomass may be changing over long time scales with climate change is important for fisheries management, especially at regional scales, as it affects our ability to make accurate near-term predictions of fisheries production. Near-term predictions of phytoplankton biomass may also benefit from knowledge of the projected magnitude of internal variability, as the chaotic nature of internal variability hampers near-term predictions (Meehl et al., 2009, 2014).

Here, we quantify changes in the internal variability (ensemble spread) of phytoplankton biomass over the next century using a large ensemble of an ESM, in which each ensemble member experiences a different phasing of internal climate variability

but is forced with a common emissions scenario. We illustrate the drivers of these changes in internal variability via statistical analysis of physical and biogeochemical model output and demonstrate their relative importance in key fisheries regions.

## 2    Methods

### 2.1    Community Earth System Model 1 Large Ensemble

#### 2.1.1    Model Description

We evaluate changes in phytoplankton biomass internal variability using output from the Community Earth System Model 1 Large Ensemble (CESM1-LE) (Kay et al., 2015). CESM1 is a fully-coupled climate model that simulates Earth's climate under historical and Representative Concentration Pathway (RCP) 8.5 external forcing by simulating the evolution of coupled atmosphere, ocean, land, and sea ice component models (Hurrell et al., 2013). The ocean physical model is the ocean component of the Community Climate System Model version 4 (Danabasoglu et al., 2012) and has a nominal 1° resolution and 60 vertical levels. The Parallel Ocean Program version 2 (POP2) ocean model consists of an upper-ocean ecological module which incorporates multi-nutrient co-limitation of nitrate, ammonium, phosphate, dissolved iron, and silicate on phytoplankton growth and dynamic iron cycling (Moore et al., 2004; Doney et al., 2006; Moore and Braucher, 2008). The Biogeochemical Elemental Cycle (BEC) model simulates three phytoplankton functional types (PFTs): diatoms, diazotrophs, and small phytoplankton (i.e., cyanobacteria, nanophytoplankton, picoeukaryotes). Each PFT plays a unique role in the marine ecosystem and occupies a distinct ecological niche. For example, diatoms grow faster in cool, high-nutrient environments while small phytoplankton thrive in warmer, low-nutrient environments. In contrast, diazotrophs are not limited by nitrogen availability due to their ability to biologically fix nitrogen from the atmosphere. Each PFT has a maximum growth rate, which is dictated by temperature (scaled by a temperature function with a Q10 of 2.0), and limited by nutrient and light availability (Moore et al., 2004, 2013). Anthropogenic warming can alter these environmental variables and, in turn, affect phytoplankton abundance and productivity. Photoadaptation (variable chlorophyll to carbon ratios) occurs in response to variations in irradiance and nutrient availability (Geider et al., 1998; Moore et al., 2004). In addition to these bottom-up controls, top-down controls, such as zooplankton grazing, can also affect phytoplankton biomass. The ecosystem model simulates a single generic zooplankton functional type (ZFT) with different grazing rates and half saturation constants prescribed for different PFTs (e.g., slower zooplankton grazing rates for larger phytoplankton i.e., diatoms). Grazing rate is computed using a Holling Type III (sigmoidal) relationship and is a function of both prey density and temperature (Figure S1, Equation 5). Zooplankton loss is a function of a linear mortality term which represents natural mortality and a non-linear predation term which represents losses from predation. Both of these loss terms scale with temperature. While zooplankton growth and loss terms both scale with temperature, a non-linear parameterization of the loss term results in a relatively larger increase in loss than increase in production with warming.

Large ensembles of ESMs are a recently developed research tool which allow us to disentangle fluctuations due to internal climate variability from those imposed by externally forced anthropogenic trends. Internal variability refers to variability in the climate system which occurs in the absence of external forcing, and includes processes related to the coupled ocean-atmosphere

system (e.g., El Niño Southern Oscillation, Pacific Decadal Oscillation) (Santer et al., 2011; Deser et al., 2010; Meehl et al., 2013). In contrast, external forcing refers to the signal imposed by processes external to the climate system, such as solar variability, volcanic eruptions, and rising greenhouse gases from fossil fuel combustion (Deser et al., 2012, 2010; Schneider and Deser, 2018). The CESM1-LE simulates the evolution of the climate system with multiple ensemble members, each initiated with slightly different atmospheric temperature fields and branched from a multi-century 1850 control simulation with constant pre-industrial forcing (Lamarque et al., 2010; Danabasoglu et al., 2012). The CESM1-LE simulates the evolution of the climate system from 1920 to 2100 with multiple ensemble members, each expressing a different phasing of internal climate variability while responding to a shared external forcing prescription (Kay et al., 2015). Variable phasing of internal climate variability (e.g., ENSO) across ensemble members can influence phytoplankton biomass variability through the propagation of physical climate variability to biologically relevant environmental variables. RCP8.5 forcing was applied from 2006 to 2100 (Meinshausen et al., 2011) with well-mixed greenhouse gases and short-lived aerosols projected by four different Integrated Assessment Models (Lamarque et al., 2010). A total of 40 ensemble members were generated for the CESM1-LE experiment. Six CESM1-LE members had corrupted ocean biogeochemistry, therefore, we use the 34 CESM1-LE members with valid ocean biogeochemistry.

### 2.1.2 Statistical Analysis of Model Output

Analyses were conducted using annual mean output at 1° resolution from 1920 to 2100. Changes in CESM1 phytoplankton internal variability can be assessed via statistical analysis of chlorophyll concentration, net primary productivity (NPP), or phytoplankton carbon concentration (an indicator of total biomass). In our analysis we focus on biomass (phytoplankton carbon concentration) because it is an important predictor variable in offline fisheries models (Christensen and Walters, 2004; Travers-Trolet et al., 2009; Lehodey et al., 2010; Maury, 2010; Blanchard et al., 2012; Christensen et al., 2015; Jennings and Collingridge, 2015; Tittensor et al., 2018). Additionally, under climate change scenarios, phytoplankton biomass may be a more reliable indicator than NPP of climate change impacts (Bopp et al., 2021). Vertical integrals (top 150m) of biomass carbon concentration from each PFT were calculated and then summed to create maps of total phytoplankton biomass.

We classified the marine environment into 11 ecologically cohesive biomes as in Tagliabue et al. (2021) and Vichi et al. (2011) (Figure S2), which are a consolidation of the 38 ecological regions defined in Longhurst (2007). The provinces were aggregated using multivariate statistical analysis of physical (i.e., salinity, temperature, mixed layer depth) and biological (i.e., chlorophyll concentration) ocean parameters to group ocean regions with similar physical and environmental conditions (Vichi et al., 2011). The ocean provinces were defined by randomly selecting from a combination of model and observational datasets and testing for statistical significance using analysis of similarities (ANOSIM) (Vichi et al., 2011). Although we consider all 11 biomes in our analysis, we analyze drivers in four biomes that are particularly relevant for fisheries production and/or of high biogeochemical interest: the subpolar Atlantic (ASP), the subarctic Pacific (SAP), the Equatorial Pacific (EQP), and the Southern Ocean (SOC) (Figure S2). ASP is a consolidation of aggregated biogeochemical provinces 4, 11, and 15, SAP a consolidation of 50 and 51, EQP a consolidation of 61, 62, and 63, and SOC a consolidation of 21, 81, 82, and 83 (Longhurst,

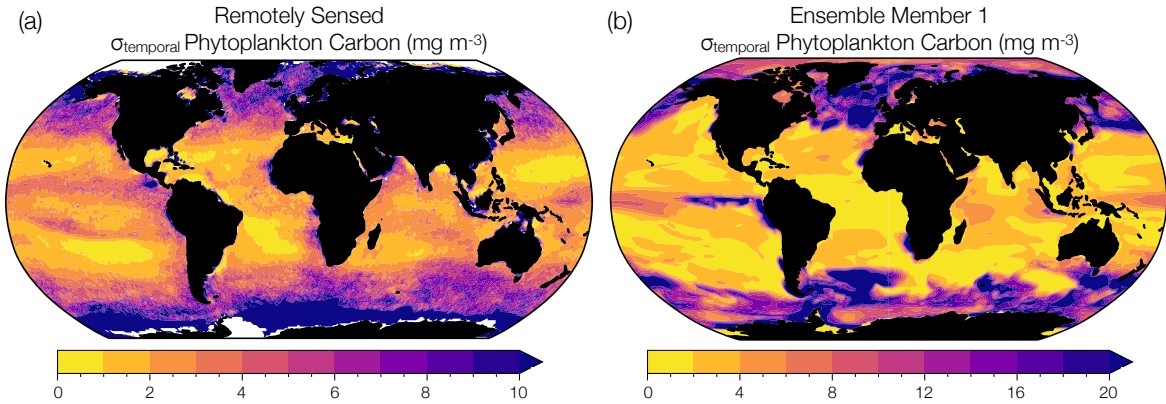

**Figure 1.** Comparison between observed and modeled phytoplankton biomass interannual variability. (a) Temporal standard deviation in annual mean phytoplankton carbon concentration (mg m$^{-3}$) reconstructed from remotely sensed chlorophyll concentrations, backscattering coefficients, and phytoplankton absorption (1998 to 2019) (Bellacicco et al., 2020). (b) Temporal standard deviation in annual mean phytoplankton carbon concentration (mg m$^{-3}$) simulated by ensemble member 1 of the CESM1-LE over the same observational period (1998 to 2019). Note the different magnitudes on the colorbars.

2007; Vichi et al., 2011) (Figure S2). Important biogeochemical regions are those characterized by coherent physical and environmental conditions, which support unique marine ecosystems and play an outsized role on global ocean biogeochemistry.

Internal variability at each location $(x,y)$ is approximated as the standard deviation $(\sigma)$ across ensemble members (EMs) at a given time $(t)$,

$$\sigma(x,y,t) = \sigma(EM(x,y,t)). \tag{1}$$

The coefficient of variation (CoV) is calculated as the standard deviation across the ensemble members divided by the ensemble mean,

$$CoV(x,y,t) = \frac{\sigma(EM(x,y,t))}{\overline{LE}^{EM}}. \tag{2}$$

The forced response of the large ensemble is calculated as the mean of ensemble members at a given location and time,

$$\overline{LE}(x,y,t) = \frac{\sum_1^n EM(x,y,t)}{n}, \tag{3}$$

where $n$ is the number of ensemble members.

We quantified the drivers of phytoplankton carbon biomass CoV in key ocean regions by generating an ensemble of boosted regression trees. Unlike linear models, boosted trees are able to capture non-linear interaction between the predictors and the

response. A regression tree ensemble is a predictive model composed of a weighted combination of multiple regression trees. At every step, the ensemble fits a new learner to the difference between the observed response and the aggregated prediction of all learners grown previously, aiming to minimize mean-squared error. We generate an ensemble of boosted regression trees (maximum tree depth = 10) using the Matlab function *fitrensemble*. Our predictor variables are the regional mean, ensemble mean temperature, mixed layer depth, incoming shortwave radiation, physically mediated iron, physically mediated phosphate, zooplankton carbon, and zooplankton grazing (diatom, small phytoplankton, or their sum), while our response variable is CoV of phytoplankton carbon (diatom, small phytoplankton, or their sum) annually resolved from 2006 to 2100. We use the Matlab function *predictorImportance* to estimate the importance of the predictors for each tree learner in the ensemble; it computes the importance of the predictors in a tree by summing changes due to splits on every predictor and dividing the sum by the total number of branches.

## 2.2 Model Evaluation

We used remotely sensed estimates of phytoplankton carbon to evaluate the representation of phytoplankton interannual variability in the CESM1-LE. In other words, we evaluate the temporal variability in modeled phytoplankton biomass from year to year. We note that this interannual variability is different than the internal variability (ensemble spread) that we discuss at length in this study, but is nevertheless a target for model validation. Although phytoplankton carbon concentrations cannot be measured directly by satellites, they can be reconstructed using algorithms that incorporate remotely sensed chlorophyll concentrations, detrital backscattering coefficients, and phytoplankton absorption (Kostadinov et al., 2016; Martinez-Vicente et al., 2017; Roy et al., 2017; Sathyendranath et al., 2020; Brewin et al., 2021). We use the observational phytoplankton carbon dataset of Bellacicco et al. (2020), annually averaged and interpolated onto a 1° grid, to evaluate interannual variability in phytoplankton biomass in a single model ensemble member. Figure 1a shows satellite derived estimates of interannual variability in phytoplankton carbon with regions of relatively low phytoplankton variability shown in yellow and regions of relatively high variability in purple. Remotely sensed observations capture areas of high interannual variability in the subpolar North Atlantic, North Pacific, and Southern Ocean and areas of low interannual variability in the subtropical gyre regions. Similar spatial patterns are apparent when compared to the range of phytoplankton interannual variability in ensemble member 1 of the CESM1-LE over the observational period (1998 to 2019) (Figure 1b). However, while the model ensemble captures regional patterns of observed variability, the CESM1-LE overestimates the magnitude of observed interannual variability. Some regions of the global ocean display a substantial mismatch in interannual variability between the model and that estimated from observations (Figure 1, Table S1). While the differences can be quite large in some regions, we note that this is an evaluation of interannual variability (rather than internal variability, the focus of this study), and that estimates from the satellite product derive from a collection of data products which may also display biases (Table S1).

As an evaluation of the model's ability to represent internal variability (ensemble spread), we compare the internal variability in chlorophyll simulated in the CESM1-LE to a synthetic ensemble generated from observed surface chlorophyll concentrations over the MODIS remote sensing record (Elsworth et al., 2020, 2021) (Figure S3; chlorophyll was readily available in the CESM1-LE and can be directly compared with our synthetic ensemble of observed surface chlorophyll). A synthetic ensemble

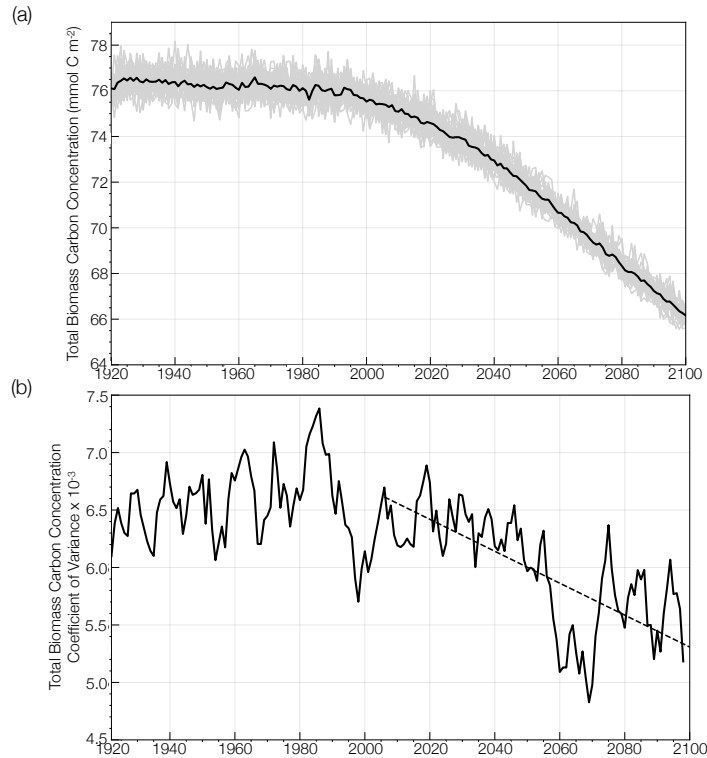

**Figure 2.** (a) Global change in annual mean total phytoplankton carbon concentration simulated by the CESM1-LE in mmol C m$^{-2}$ from the historical period through the RCP8.5 forcing scenario (1920 to 2100). The ensemble mean is shown in the black curve and the 34 individual ensemble members are shown in the gray curves. (b) Global change in the coefficient of variation in annual mean total phytoplankton carbon concentration over the same period, smoothed using a 5 year window. Trend in the coefficient of variation over the RCP8.5 forcing scenario is shown in the black dashed line.

is a technique that allows the observational record to be statistically emulated to create multiple possible evolutions of the observed record, each with a unique sampling of internal climate variability (McKinnon et al., 2017; McKinnon and Deser, 2018). Compared to the internal variability over the observational period (2002 to 2020) (purple circle, (Figure S3), the model ensemble slightly overestimates the magnitude of internal variability in chlorophyll observed in the real world.

Taken together, our model validation exercises demonstrate that the model tends to overestimate both the temporal (interannual) variability and the internal variability in phytoplankton, as compared to satellite observations on both global and regional scales. Thus, we must interpret our findings with this caveat in mind.

## 3   Results

We evaluate the change in mean phytoplankton biomass and its internal variability across the CESM1-LE globally and region-ally. Annually averaged, global mean, upper-ocean (top 150m) integrated phytoplankton biomass across the model ensemble decreases from 76.1 mmol C m$^{-2}$ to 66.2 mmol C m$^{-2}$ from the historical period through the RCP8.5 forcing scenario (1920 to 2100), a decline of 13% (black curve; Figure 2a). The change in the mean is calculated as the difference between the first (1920 to 1930) and last (2090 to 2100) decades across the historical and RCP8.5 forcing scenario. Phytoplankton biomass declines globally, except in polar regions (Figure 3a). Regional changes in mean phytoplankton biomass across the RCP8.5 forcing scenario (2006 to 2100) display increasing biomass in portions of the Arctic and the Southern Ocean that gradually become ice-free over the century (on the order of 20-40% of the mean biomass across the century) and decreasing biomass across the subtropical gyres (on the order of 15-30% of the mean biomass across the century; Figures 3a, S4a). In the North Atlantic subpolar gyre, the phytoplankton biomass declines by 40-50% of its mean (Figures 3a, S4a). This result is consistent with previous modelling studies which identified a 50% reduction in North Atlantic primary production associated with AMOC weakening during the last glacial period (Schmittner, 2005). A weakening of the AMOC is also projected with anthropogenic warming (Manabe and Ronald, 1993; Stocker and Schmittner, 1997).

Regional changes in phytoplankton biomass are dominated by changes in diatom and small phytoplankton (Table 1). We aggregate biomass across 11 ecological provinces (Vichi et al., 2011; Tagliabue et al., 2021), and present changes in total and PFT biomass over the RCP8.5 scenario in Table 1. The CESM1-LE simulates the largest decline in total phytoplankton carbon concentration in the Atlantic subpolar (ASP) region, where diatom biomass declines by ∼80 mmol C m$^{-2}$, and small phytoplankton biomass increases slightly (∼8 mmol C m$^{-2}$). We observe moderate decreases in the subpolar Pacific (SAP) region that are again driven by declines in diatom carbon concentration, with minor contributions from changes in small phytoplankton carbon concentration (Table 1). The CESM1-LE simulates a smaller decline in total carbon concentration in the Equatorial Pacific (EQP) region, where diatom biomass declines ∼7 mmol C m$^{-2}$ and small phytoplankton biomass declines ∼5 mmol C m$^{-2}$. The smallest decline in total carbon concentration occurs in the South Pacific subtropical gyre (SPS) region, where diatom biomass declines ∼4.3 mmol C m$^{-2}$ and small phytoplankton biomass declines ∼4.6 mmol C m$^{-2}$.

Internal variability in global phytoplankton biomass, which is indicated by the spread across the individual ensemble mem-bers (gray lines; Figure 2a), declines over the RCP8.5 forcing period from 2006 to 2100. To quantify how the range of internal variability in phytoplankton biomass is changing with anthropogenic warming, we calculated the coefficient of variation as the standard deviation across the ensemble members for a given year (ensemble spread) divided by the ensemble mean. Figure 2b illustrates the change in the coefficient of variation from the historical period through the RCP8.5 forcing scenario (1920 to 2100). The coefficient of variation is relatively constant across the historical period (1920 to 2005), and then significantly declines by ∼20% from 2006-2100.

A decrease in global phytoplankton internal variability with anthropogenic warming is not unique to the CESM1-LE. We illustrate this by analyzing surface phytoplankton chlorophyll (rather than biomass; surface chlorophyll was readily available in the CMIP5 archive) from three other CMIP5 ESM large ensembles which include representation of ocean biogeochemistry:

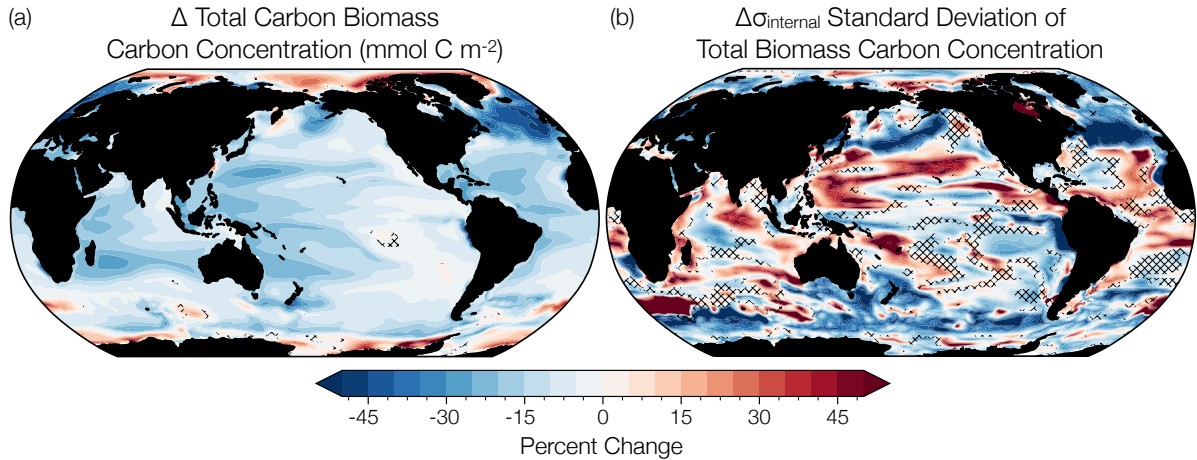

**(a)** Δ Total Carbon Biomass
Carbon Concentration (mmol C m⁻²)

**(b)** Δσ$_{internal}$ Standard Deviation of
Total Biomass Carbon Concentration

Percent Change

**Figure 3.** (a) Percentage change in annual total phytoplankton carbon concentration over the RCP8.5 forcing scenario (2006 to 2100) simulated by the CESM1-LE. (b) Percentage change in annual total phytoplankton internal variablity over the same period. The change in the mean and the variability are calculated using averages across the first (2006 to 2016) and last (2090 to 2100) decades of the RCP8.5 forcing scenario. Hatched areas indicate regions of trend insignificance determined by a t-test with a p value greater than 0.05. Summary statistics for the t-test are available in the supplemental information (Table S2).

the GFDL-ESM2M from the Geophysical Fluid Dynamics Laboratory (GFDL; Dunne et al., 2012, 2013), the CanESM2 from the Canadian Centre for Climate Modelling and Analysis (Christian et al., 2010; Arora et al., 2011), and the MPI-ESM-LR from the Max Planck Institute (MPI; Giorgetta et al., 2013; Ilyina et al., 2013), consisting of 30, 50, and 100 ensemble members, respectively. Similarly to the CESM1-LE, historical forcing was applied through 2005, followed by RCP8.5 forcing through 2100. While there is substantial spread in the mean coefficient of variation across the four models, a similar decline

in the coefficient of variation can be observed across each of the four ESM ensembles, (Figure S3). From 2006 to 2100, the coefficient of variation decreases by $3.3 \times 10^{-5}$ yr$^{-1}$ in the CESM1-LE, $2.0 \times 10^{-4}$ yr$^{-1}$ in the MPI-ESM-LR1, $5.2 \times 10^{-5}$ yr$^{-1}$ in the CanESM2, and $3.9 \times 10^{-4}$ yr$^{-1}$ in the GFDL-ESM2M. The change in the coefficient of variation is calculated using averages across the first (2006 to 2016) and last (2090 to 2100) decades of the RCP8.5 forcing scenario. These declines are statistically significant in all model ensembles with the exception of the MPI-ESM-LR1 (Figure S3).

In comparison to the mean change in phytoplankton biomass, changes in phytoplankton internal variability with time are spatially more heterogeneous across the global ocean (Figure 3b). The largest decreases in internal variability are apparent in the North Atlantic and North Pacific subpolar regions (on the order of 50-70% of the mean biomass internal variability), with smaller declines in the Equatorial Pacific and Southern Oceans (on the order of 30-50% of the mean biomass internal variability) (Figure 3b, S4b). Changes in internal variability in the subtropical regions are characterized by mixed trends, with

areas of both increasing and decreasing internal variability across the RCP8.5 forcing scenario (Figure 3b, S4b).

Global changes in total phytoplankton biomass standard deviation are a manifestation of changes in diatom and small phytoplankton variability (Table 1). We observe the largest magnitude decline in total phytoplankton carbon standard deviation in the subpolar Atlantic (ASP) region, where diatom standard deviation declines by $\sim$10 mmol C m$^{-2}$ and small phytoplankton standard deviation declines by $\sim$2 mmol C m$^{-2}$ (Table 1). The CESM1-LE simulates a moderate magnitude decline in total

phytoplankton standard deviation in the subarctic Pacific (SAP) region, driven by a decrease in small phytoplankton standard deviation ($\sim$2 mmol C m$^{-2}$) with minor contributions from declines in diatom standard deviation ($\sim$1 mmol C m$^{-2}$) (Table 1). Moderate declines in standard deviation are also simulated in the Arctic (ARC), North Atlantic subtropical gyre (NAS), Southern Ocean (SOC), and Equatorial Pacific (EQP) regions, driven by declines in diatom carbon standard deviation in the SOC region and declines in small phytoplankton internal variability in the EQP region (Table 1).

**Table 1.** Changes in phytoplankton biomass and its internal variability in the CESM1-LE from 2006 to 2100 for the 11 ecological provinces defined in Vichi et al. (2011) and Tagliabue et al. (2021). Units are mmol C m$^{-2}$. The change in the mean and standard deviation are calculated using averages across the first (2006 to 2016) and last (2090 to 2100) decades of the RCP8.5 forcing scenario.

| | Region | Change in Mean | | | Change in Standard Deviation | | |
|---|---|---|---|---|---|---|---|
| Biome | Name | Total | Diatom | Small | Total | Diatom | Small |
| ARC | Arctic | –21 | –58 | +37 | –1.4 | –2.8 | –0.3 |
| ASP | Atlantic subpolar | –71 | –79 | +8.2 | –5.6 | –9.9 | –2.2 |
| NAS | North Atlantic subtropical gyre | –18 | –15 | –2.9 | –1.8 | –2.8 | –0.3 |
| EQA | Equatorial Atlantic | –12 | –6.6 | –5.9 | –0.1 | –0.4 | +0.2 |
| SAS | South Atlantic subtropical gyre | –10 | –7.2 | –3.1 | –0.5 | –0.6 | –0.1 |
| IND | Indian Ocean | –11 | –6.1 | –4.7 | +0.1 | 0 | +0.1 |
| SAP | subarctic Pacific | –21 | –15 | –5.4 | –0.1 | –1.4 | –2.4 |
| NPS | North Pacific subtropical gyre | –11 | –5.6 | –4.9 | –0.2 | –0.4 | +0.1 |
| EQP | Equatorial Pacific | –12 | –6.6 | –5.0 | –2.0 | –2.0 | –0.2 |
| SPS | South Pacific subtropical gyre | –8.9 | –4.3 | –4.6 | –0.1 | 0 | –0.1 |
| SOC | Southern Ocean | –9.3 | –2.8 | –6.6 | –1.0 | 0 | –1.3 |

To guide our analysis of changing phytoplankton biomass internal variability, we considered the dominant ecological assemblage across different regions of the global ocean. The CESM1-LE simulates three phytoplankton functional types, each of which thrive in distinct regions of the global ocean. Diatoms dominate in the subpolar Atlantic and Pacific, the Eastern Equatorial upwelling zone, and portions of the Southern Ocean, while small phytoplankton dominate across the subtropical gyres and portions of the Southern Ocean (Figure 4). In contrast, diazotrophs, a minor contributor to total carbon biomass, are present at

such low concentrations that they do not dominate anywhere in the global ocean (Figure 4). Using the ecologically cohesive regions defined by Tagliabue et al. (2021) and Vichi et al. (2011), we selected areas that align with the most productive fisheries regions by catch in the Atlantic and Pacific basins (FAO, 2020), as well as regions of global biogeochemical importance for further analysis. In each ecological region we identified the dominant phytoplankton functional type to include in our analysis.

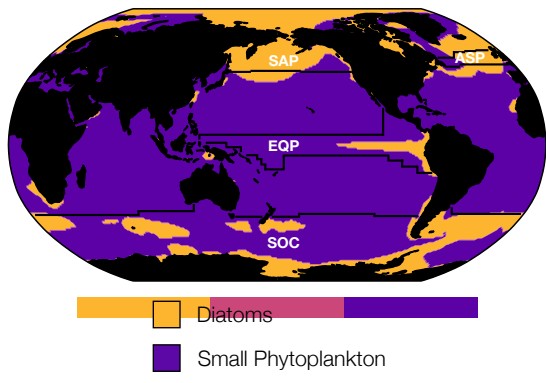

**Figure 4.** Distribution of the dominant phytoplankton functional type in biomass carbon averaged across the RCP8.5 forcing scenario (2006 to 2100). The CESM1-LE simulates three phytoplankton functional types: diatoms, diazotrophs, and small phytoplankton. Regions where diatoms dominate are shown in yellow and regions where small phytoplankton dominate are shown in purple. Diazotrophs do not dominate in any region of the global ocean. The four ecological provinces are shown: subpolar Pacific (SAP), subpolar Atlantic (ASP), Equatorial Pacific (EQP), and Southern Ocean (SOC).

In regions where multiple phytoplankton functional types dominated, we used total carbon concentrations to reflect the mixed ecological assemblage.

We identify the importance of different predictors to changing phytoplankton biomass CoV in four distinct ecological regions using a machine learning (boosted regression tree) approach. In the subpolar Atlantic (ASP) and subpolar Pacific (SAP) ecological provinces (Figure 4), diatom biomass CoV declines between the beginning and end of the century (Table 1). In the Atlantic subpolar region, the most important predictor of diatom biomass CoV is phosphate advection, with smaller contributions from zooplankton carbon (Figure 5a). In the subarctic Pacific region, sea surface temperature is the most important predictor of diatom biomass CoV, with phosphate advection playing a secondary role (Figure 5b).

As the Southern Ocean (SOC) and Equatorial Pacific (EQP) ecological provinces are characterized by mixed phytoplankton assemblages where both diatoms and small phytoplankton dominate (Figure 4), we identify the predictors of total phytoplankton CoV here. In contrast to the subpolar Atlantic and subpolar Pacific provinces, we observe a relatively smaller decline in phytoplankton CoV between the beginning and end of the century in the Southern Ocean (Table 1). The most important predictors of phytoplankton CoV in the Southern Ocean (SOC) region derive from solar flux, with more minor contributions from iron and phosphate advection (Figure 5c). In the Equatorial Pacific region, zooplankton carbon is the most important predictor of total phytoplankton CoV, while iron and phosphate advection play less of a predictive role (Figure 5d).

In all four ecological provinces, a combination of bottom-up controls (e.g., nutrient supply, light availability) and top-down controls (e.g., grazer biomass) predict the decline in phytoplankton biomass CoV with anthropogenic warming. Our statistical analysis reveals that phosphate advection is an important predictor in the high-latitude regions of both the subpolar Atlantic

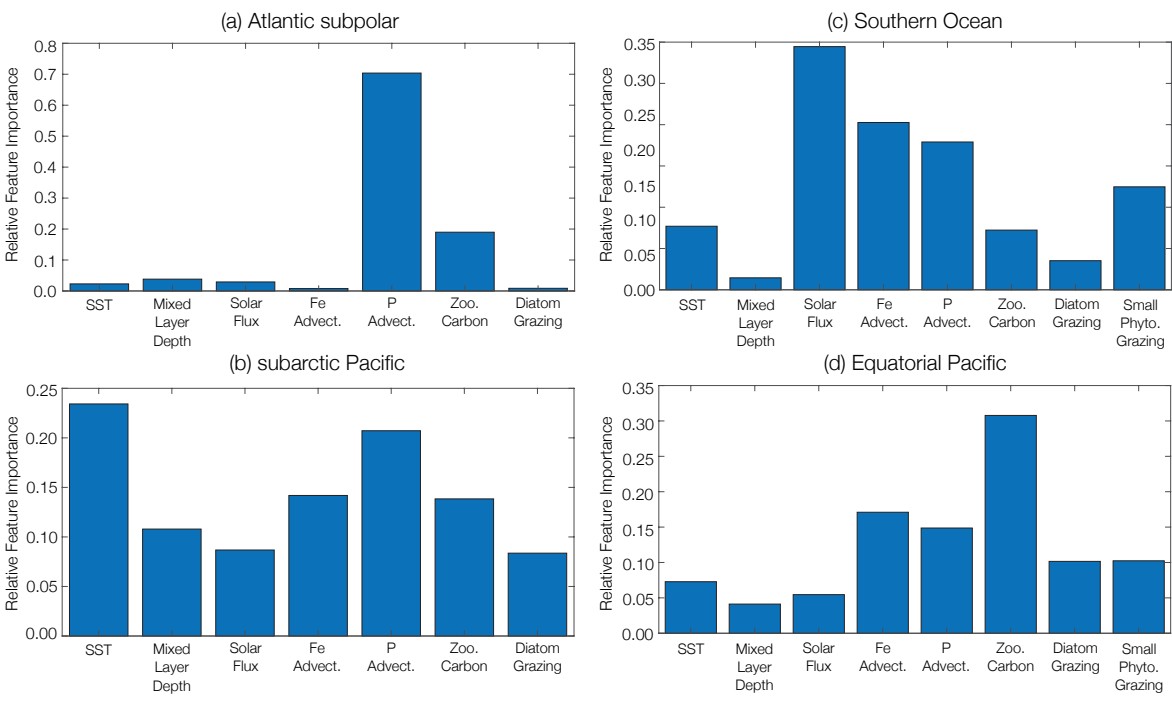

**Figure 5.** Relative importance of predictor variables on phytoplankton biomass coefficient of variation across the RCP8.5 forcing scenario (2006 to 2100). Marine ecological regions are defined in Tagliabue et al. (2021). Regions were selected which aligned with the highest fisheries catch in the (a) Atlantic and (b) Pacific basins and the biogeochemically important (c) Southern Ocean and (d) Equatorial Pacific regions. The dominant phytoplankton functional type is considered in each region. In regions with a mixed ecological assemblage, total phytoplankton carbon is considered.

and Pacific, with sea surface temperature playing an important role in the subpolar Pacific. However, in the Southern Ocean and the Equatorial Pacific, solar flux and grazer biomass dominate the predictive skill in phytoplankton biomass CoV.

## 4 Conclusions and Discussion

We quantify both global and regional changes in phytoplankton internal variability across the RCP8.5, or business-as-usual forcing scenario in the CESM1-LE. We observe a global decline in phytoplankton internal variability in the model ensemble, which is reflected in similar declines in phytoplankton internal variability across a suite of CMIP5 models (Figure S3). Regional changes in phytoplankton variability with anthropogenic climate change in the model ensemble are spatially heterogeneous, with highly productive fisheries regions and important global biogeochemical regions experiencing large changes in internal

variability. Using a machine learning approach, we identify the importance of different predictors to changing phytoplankton

biomass internal variability. In all four ecological provinces, a combination of bottom-up controls (e.g., nutrient supply, light availability) and top-down controls (e.g., grazer biomass) predict the decline in phytoplankton biomass CoV with anthropogenic warming.

While the CESM1-LE represents the overall spatial pattern of observed interannual variability in phytoplankton carbon, the model overestimates the magnitude of observed interannual and internal variability in phytoplankton on regional scales. This caveat is particularly important to consider when interpreting projections from offline fisheries models in the context of fisheries adaptation and planning in a warming climate.

Our statistical analysis approach has inherent limitations, especially in the context of a attributing changes in an inherently coupled system (i.e., one in which predictor variables co-vary). In a coupled system such as this, it is difficult to definitively identify cause and effect. In this context, the statistical method can be used as an effective tool to provide a first-order approximation of contributions to changing phytoplankton CoV.

While many studies attribute bottom-up controls to changing phytoplankton with anthropogenic warming (Steinacher et al., 2010; Bopp et al., 2013; Lotze et al., 2019; Tittensor et al., 2021), top-down controls may also play an important role, particularly in our understanding of changing phytoplankton biomass and its internal variability. Our study demonstrates a connection between phytoplankton internal variability and zooplankton carbon in the subpolar North Pacific and Equatorial Pacific. Previous studies of phytoplankton change with climatic warming have demonstrated that grazing pressure is a contributor to biomass decline in low to intermediate latitude regions across a suite of model simulations with different marine ecosystem models (Laufkötter et al., 2015) and that top-down controls can affect regional changes in NPP and export production (Bopp et al., 2001) and is a contributor to future shifts in bloom timing ((Yamaguchi et al., 2022)). While grazing pressure has been shown to increase in response to climate change, several ecosystem models have also identified zooplankton grazing as a dominant contributor to phytoplankton assemblage succession during blooms (Hashioka et al., 2012; Prowe et al., 2012a). Additionally, top-down controls have also been observed to affect the onset of the spring bloom (Behrenfeld, 2010; Behrenfeld et al., 2013), to influence primary production in a trait-based ecosystem model (Prowe et al., 2012b).

The relative simplicity of the ocean biogeochemical ecosystem model in CESM1 (e.g., representation of a single zooplankton functional type with multiple grazing rates) may limit a more detailed evaluation of changing grazing pressures with climate change. While the recent parameterization of the biogeochemical ecosystem model in CESM2 (MARBL) includes similar representation of three PFTs and a single adaptive ZFT (Long et al., 2021), more complex configurations of MARBL include explicit representation of additional PFTs such as coccolithophores (Krumhardt et al., 2019) and ZFTs. Additional insights into contributions to internal variability may be gained using more complex models. Additionally, the use of an ecosystem model of higher complexity may provide more realistic projections of the marine ecosystem with climate change considering change in phytoplankton and zooplankton species diversity with anthropogenic warming (Benedetti et al., 2021).

The magnitude and direction of regional changes in phytoplankton internal variability are an essential constraint for near-term (subseasonal to decadal) predictions of the local marine ecosystem, particularly in important fisheries regions such as the subpolar Atlantic (ASP) and the subpolar Pacific (SAP) ecological provinces (FAO, 2020). Accurate near-term predictions require foreknowledge of both internal climate variability and external climate change signals. On subseasonal to decadal

timescales, the magnitude of internal climate variability is often stronger than forced climate change signals (Meehl et al., 2009, 2014). In this context, a decline in phytoplankton internal variability with anthropogenic climate change may improve the accuracy of near-term predictions of phytoplankton biomass, producing more reliable forecasts of fisheries productivity and marine carbon cycling. Future work can utilize these constraints on phytoplankton internal variability, particularly on regional
scales, to inform climate mitigation and adaptation efforts.

*Data availability.*  CESM1-LE output is available from the Earth System Grid at http://www.cesm.ucar.edu/projects/community-projects /LENS/data-sets.html.

*Author contributions.*  GE and NL conceptualized the study. GE analyzed simulation results and prepared the manuscript. N.L. assisted in preparing the manuscript. KK assisted in analysis of simulation results. TM, KK, and SS assisted in reviewing the manuscript.

*Competing interests.*  We declare no competing interests.

*Acknowledgements.*  Computational resources were provided by the Computational and Information Systems Laboratory (CISL) at the National Center for Atmospheric Research (NCAR), through a resource allocation to GWE and NSL. GWE and NSL are grateful for support from the National Science Foundation (NSF) (OCE-1558225, OCE-1752724).

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

In our discussion of zooplankton grazing as a contributor to changing phytoplankton internal variability with anthropogenic warming, we consider the parameterization of zooplankton grazing in the CESM1-LE. The biogeochemical ecosystem model simulates a single generic zooplankton functional type (ZFT) with different grazing rates and half saturation constants pre-scribed for different PFTs (e.g. slower zooplankton grazing rates for larger phytoplankton). Grazing rate for the single ZFT is computed using a Holling Type III (sigmoidal) relationship:

$$G = g_{max} \cdot T_{lim} \cdot Z \cdot \frac{P^2}{P^2 + K^2} \tag{4}$$

where $g_{max}$ is the maximum grazing rate, $T_{lim}$ is the temperature limitation (Q10) function, Z is the zooplankton concentration, P is the phytoplankton concentration, and K is the half-saturation constant for grazing. Zooplankton loss scales with temperature and a linear mortality term which represents zooplankton losses from predation.

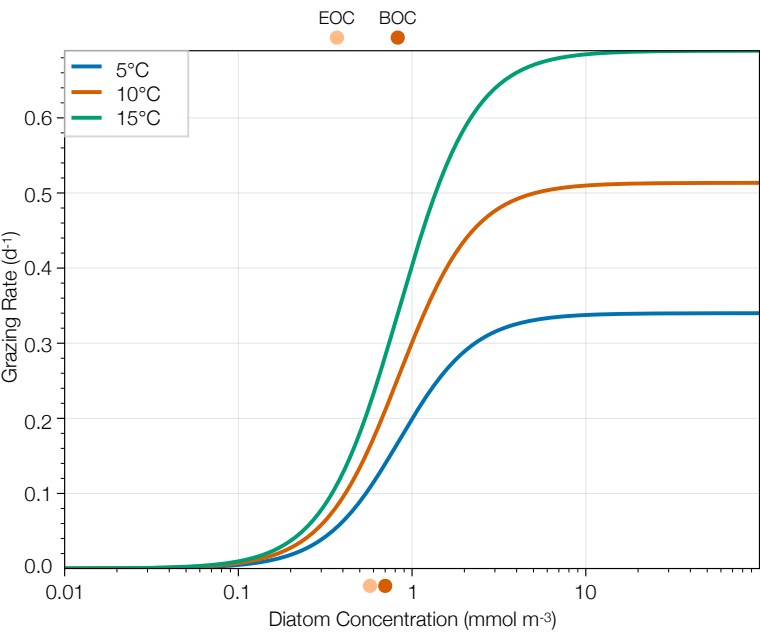

**Figure S1.** Holling Type III (sigmoidal) functional parameterization of zooplankton grazing rate in the biogeochemical ecosystem model of the CESM1-LE across a range of temperatures. Changes in diatom concentration between the beginning and end of the century (BOC and EOC, respectively) are shown in the dark and light orange circles, respectively, with the changes in the ASP region shown above and changes in the SAP region shown below.

Figure S1 illustrates changes in grazing rate as a function of diatom concentration using this parameterization. To approxi-mate the effects of climatic warming, we plot the relationship for across a series of increasing temperatures: (blue) 5°C, (orange)

10°C, and (green) 15°C. The maximum grazing rate increases with warming temperatures. Changes in diatom concentration in mmol m$^{-3}$ between the beginning and end of the century are denoted by dark and light orange circles, respectively.

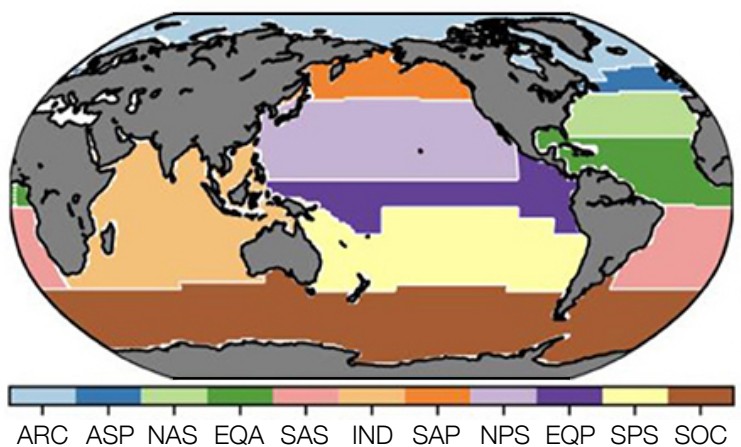

**Figure S2.** The 11 ocean ecological provinces defined in Tagliabue et al. (2021) and Vichi et al. (2011). Provinces were aggregated using multivariate statistical analysis of physical (i.e., salinity, temperature, mixed layer depth) and biological (i.e., chlorophyll concentration) ocean parameters to group ocean regions with similar physical and environmental conditions. Figure adapted from Tagliabue et al. (2021).

**Table S1.** The temporal standard deviation of phytoplankton biomass ($\sigma_{temporal}$) for ensemble member 1 of the CESM1-LE and the satellite observations from 1998 to 2019 averaged across the 11 ecological provinces defined in Vichi et al. (2011) and Tagliabue et al. (2021). Units are mg C m$^{-3}$.

| Biome | Name | $\sigma_{temporal,model}$ | $\sigma_{temporal,obs}$ |
|-------|------|-----------|-----------|
| ARC | Arctic | 2.7 | 4.5 |
| ASP | Atlantic subpolar | 9.7 | 4.1 |
| NAS | North Atlantic subtropical gyre | 2.8 | 1.7 |
| EQA | Equatorial Atlantic | 1.3 | 1.4 |
| SAS | South Atlantic subtropical gyre | 1.1 | 1.2 |
| IND | Indian Ocean | 0.81 | 2.0 |
| SAP | subarctic Pacific | 3.7 | 4.0 |
| NPS | North Pacific subtropical gyre | 0.85 | 1.5 |
| EQP | Equatorial Pacific | 5.8 | 1.8 |
| SPS | South Pacific subtropical gyre | 0.60 | 0.93 |
| SOC | Southern Ocean | 2.7 | 2.7 |

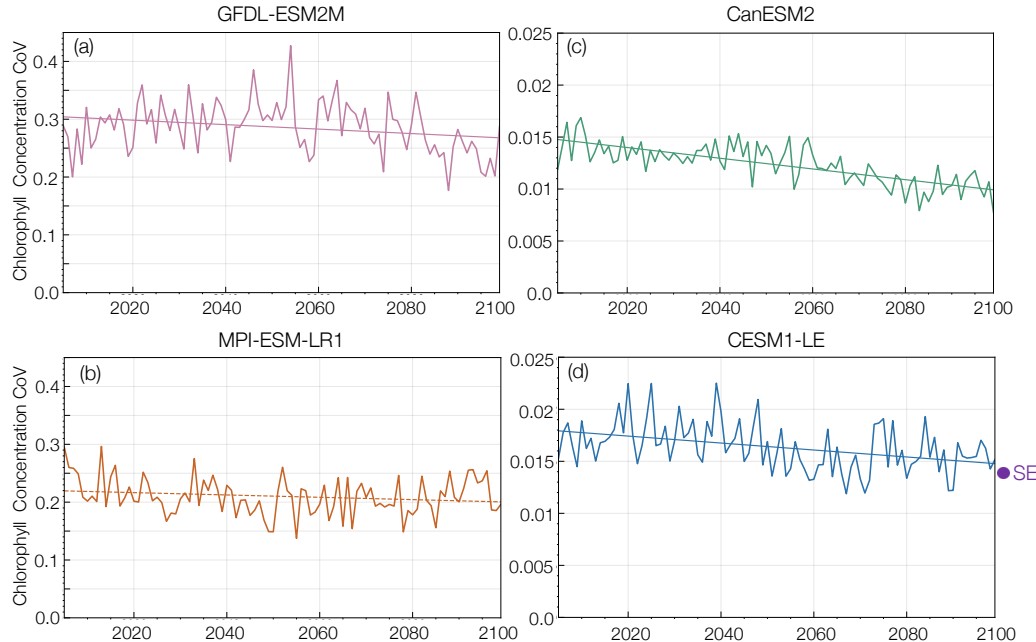

**Figure S3.** Coefficient of variation (internal standard deviation divided by ensemble mean) in annual mean global surface ocean chlorophyll concentration from 2006 to 2100 across a suite of CMIP5 model ensembles: (a) (pink) GFDL-ESM2M (b) (orange) MPI-ESM-LR1 (c) (green) CanESM2 (d) (blue) CESM1-LE. The average coefficient of variation of the synthetic ensemble (SE) created using the MODIS surface ocean chlorophyll record is shown in the purple dot on the vertical axis (Elsworth et al., 2020, 2021). Trend significance is determined by a t-test with a p-value less than 0.05.

To provide context for the CESM1-LE results, we examine changes in chlorophyll internal variability from a subset of the Coupled Model Intercomparison Project 5 (CMIP5) models (Taylor et al., 2011): the GFDL-ESM2M from the Geophysical Fluid Dynamics Laboratory (GFDL; (Dunne et al., 2012, 2013), the CanESM2 from the Canadian Centre for Climate Modelling and Analysis (Christian et al., 2010; Arora et al., 2011), and the MPI-ESM-LR from the Max Planck Institute (MPI; (Giorgetta et al., 2013; Ilyina et al., 2013), consisting of 30, 50, and 100 ensemble members, respectively. Similarly to the CESM1-LE, historical forcing was applied through 2005, followed by RCP8.5 forcing through 2100.

We compare the internal variability in chlorophyll observed among the large ensembles to a synthetic ensemble generated from observational chlorophyll concentrations over the MODIS remote sensing record (Elsworth et al., 2020, 2021). A synthetic ensemble is a technique that allows the observational record to be statistically emulated to create multiple possible evolutions of the observed record, each with a unique sampling of internal climate variability (McKinnon et al., 2017; McKinnon and Deser, 2018). We use the synthetic ensemble of chlorophyll concentration to compare the variability observed in the real world to the variability simulated across a suite of ESM ensembles.

**Table S2.** Summary statistics for the t-test performed on total phytoplankton biomass to determine trend significance across the RCP8.5 forcing scenario (2006 to 2100). Datasets are normally distributed.

| Variable | Sample Size | Mean | Standard Error | 95% CI |
|---|---|---|---|---|
| Total Phytoplankton Biomass Mean Trend | 94 | -0.0697 | 0.00459 | -0.0743 to -0.0651 |
| Total Phytoplankton Biomass Standard Deviation Trend | 94 | -0.0164 | 0.00323 | -0.0196 to -0.0132 |

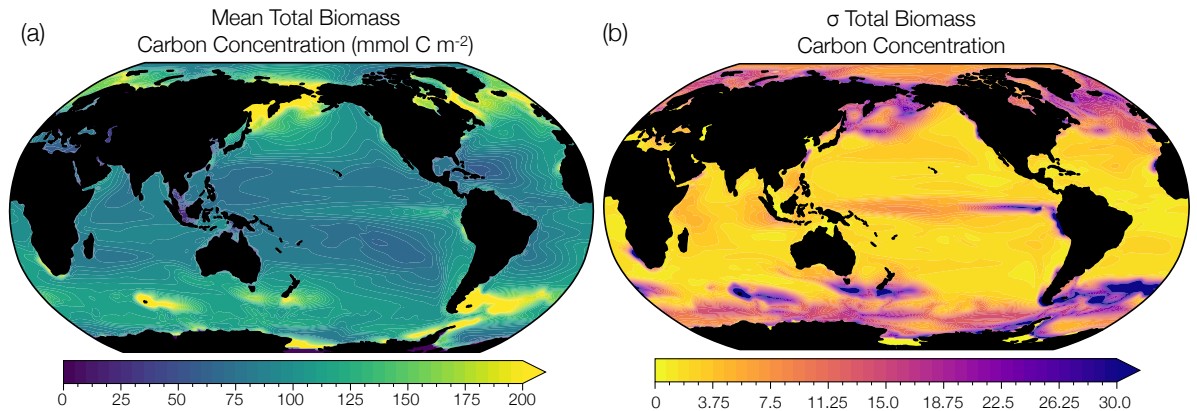

**Figure S4.** (a) Total phytoplankton carbon concentration simulated by the CESM1-LE in mmol C m$^{-2}$ averaged across the RCP8.5 forcing scenario (2006 to 2100). (b) Internal standard deviation in total phytoplankton carbon concentration averaged over the same period. The change in the coefficient of variation is calculated using averages across the first (2006 to 2016) and last (2090 to 2100) decades of the RCP8.5 forcing scenario.

To provide context for Figure 3, we include the spatial distribution of total phytoplankton carbon concentration (Figure S4a) and internal standard deviation in phytoplankton carbon concentration (Figure S4b) simulated by the CESM1-LE across the RCP8.5 forcing scenario (2006 to 2100). Total phytoplankton carbon concentration is relatively high in the subpolar Atlantic and Pacific, the Southern Ocean, and the Eastern Equatorial Upwelling Zone and relatively low in the subtropical gyre regions (Figure S4a). Regions of relatively high phytoplankton carbon concentrations correspond to regions of high internal variability
(Figure S4b).