# Peer review of "Anthropogenic climate change drives non-stationary phytoplankton internal variability"

_EGUsphere, 2022_

## Author Comment (AC1)

Review 1 of Elsworth et al. (2022) "Anthropogenic climate change drives non-stationary phytoplankton variance", submitted to Biogeosciences.

The authors investigated future changes in interannual variability of phytoplankton carbon biomass by using the CESM1 large ensemble simulation. Their results seem to indicate highly spatially heterogeneous response of interannual variability in the biomass to the global warming by the end of the 21$^{st}$ century and relatively important contribution from changes in "top-down" control of phytoplankton growth.

I totally agree with the authors' initial point that, especially in the context of ocean biogeochemistry, future changes in variabilities have not been paid much attentions compared to those in the climate mean states, although these are critically important on decisions of mitigation and adaptation policy. I don't think that the study has no potential for being a step to help our understanding about the ocean ecosystem (including from lower to higher trophic levels) response to climate changes. However, I can not recommend the editor to publish the current manuscript to BioGeosciences, because of the following two concerns: (1) Model validity and (2) authors' conceptual misunderstanding about MLR analysis.

(1) Model validity: The model ability to represent observed variability is critical on judging if projected future changes are valid. The author must show 1) "additional model-observation comparisons" with choosing the variables which are relevant to this study's focus (i.e., phytoplankton biomass) and 2) "evidence" on which results projected from the model with biases can be considered conclusive.

The authors showed the model-observation comparison of variability of annual mean phytoplankton carbon biomass (main target of this study) in Figure 1 and mentioned "Similar spatial patterns (to observations) are apparent (in the model)" in L139. But, for me, obviously, the model special pattern has different spatial characteristics from the observations. In the high latitudes, the observation shows the maximum variance in the most pole-side latitudes in the both hemispheres, while the model shows the variance maximum in somehow equator latitude around 50-50N and 50-60S. In the equator, although there is a strong latitudinal maximum along the equator in the model, no such structure can be seen in the observation, rather higher variabilities are observed in the off-equatorial regions. Moreover, model overestimations of the observed variability can exceed 200% in the equator and the subpolar North Atlantic.

The author also showed the model validity by comparing global internal variance in chlorophyll between the model ensemble and the observational ensemble (Figure S2). However, this study's focus is the phytoplankton carbon biomass, not chlorophyll, and these two can have very different spectrums. I think that model-observation comparisons in the biomass are more suitable for the purpose and the author should assess the model in the regional scale (not global), given the spatial heterogeneous response of the biomass.

We agree that a more ideal model validation study would include an evaluation of the modeled internal variability in phytoplankton biomass as compared to that of observations. Unfortunately, such an evaluation is not possible with the tools we have on hand. As such, we opted to evaluate the modeled phytoplankton variability using two approaches: (a) assess the temporal variability in

modeled ocean phytoplankton carbon biomass as compared to that inferred from satellite observations of chlorophyll, backscattering coefficients, and phytoplankton absorption, and (b) assess the internal variability (ensemble spread) in modeled phytoplankton using a synthetic ensemble of global ocean chlorophyll concentrations derived from satellites. While neither is the perfect evaluation tool, when taken together, they provide a holistic view of modeled variability as compared to observed variability in phytoplankton. We agree with the reviewer with regard to their point about (a) – indeed, there are some regions in the ocean where there is a substantial mismatch in the temporal variability represented by the model and that estimated from observations (Figure 1). In response to this comment, we now include and make reference to a supplemental table quantifying the temporal standard deviation in each of the 11 ecological provinces (Table S1). While the differences can be quite large in some regions, we note that this is an evaluation of temporal variability (not internal variability, the focus of our paper results), and that the estimates from the satellites are derived from a collection of data products and may also have biases. We include a new paragraph in the methods section describing these caveats.

| Biome | Name | $\sigma_{temporal,model}$ | $\sigma_{temporal,obs}$ |
|-------|------|------|------|
| ARC | Arctic | 2.7 | 4.5 |
| ASP | Atlantic subpolar | 9.7 | 4.1 |
| NAS | North Atlantic subtropical gyre | 2.8 | 1.7 |
| EQA | Equatorial Atlantic | 1.3 | 1.4 |
| SAS | South Atlantic subtropical gyre | 1.1 | 1.2 |
| IND | Indian Ocean | 0.81 | 2.0 |
| SAP | subarctic Pacific | 3.7 | 4.0 |
| NPS | North Pacific subtropical gyre | 0.85 | 1.5 |
| EQP | Equatorial Pacific | 5.8 | 1.8 |
| SPS | South Pacific subtropical gyre | 0.60 | 0.93 |
| SOC | Southern Ocean | 2.7 | 2.7 |

Table S1: " The temporal standard deviation of phytoplankton biomass ($\sigma_{temporal}$) for ensemble member 1 of the CESM1-LE and the satellite observations (Bellacicco et al., 2020) from 1998 to 2019 averaged across the 11 ecological provinces defined in Vichi et al. (2011) and Tagliabue et al. (2021). Units are mg m$^{-3}$."

Line 163: "Some regions of the global ocean display a substantial mismatch in temporal variability between the model and that estimated from observations (Figure 1, Table S1). While the differences can be quite large in some regions, we note that this is an evaluation of temporal variability (rather than internal variability, the focus of this study), and that estimates from the satellite product derive from a collection of data products which may also display biases (Table S1)."

Line 176: "Taken together, our model validation exercises demonstrate that the model tends to overestimate both the temporal (interannual) variability and the internal variability in phytoplankton as compared to satellite observations on both global and regional scales. Thus, we must interpret our findings with this caveat in mind. The change in variance that we model is likely an upper-end estimate."

Line 287: "While the CESM1-LE represents the overall spatial pattern of observed interannual variability in phytoplankton carbon, the model overestimates the magnitude of observed interannual and internal variance in phytoplankton on regional scales…"

(2) MLR analysis: The methodology is unclear and it seems wrong.

The author tried to reconstruct the contribution of each driver variable to phytoplankton biomass using the MLR coefficients (equation 3 and 4). However, it obviously failed. As shown in Figure S6, the reconstructed "Total Carbon" is not equal to the sum of the other terms (i.e., equation 3 and 4 are not correct), maybe because of inaccurate MLR coefficients, neglecting offset term or strong multicollinearity between variables (e.g., MLD and SST, SST and Solar).

Linear decompositions should be applied for "change/anomaly", not for "climatology (10-year mean)."

Given a function F(X,Y,Z), in general, the first order Taylor expansion is robust only for a small change in the F ($\Delta F$),

$$\Delta F = (\partial F/\partial X)\Delta X+(\partial F/\partial Y)\Delta Y+(\partial F/\partial Z)\Delta Z + \text{(Residuals from high-order and cross terms)}.$$

The author should apply such analysis for "change" (not "climatology") considering residual terms. As the authors also mentioned, the partial differential coefficients are time-varying. The authors should be able to calculate the coefficients analytically using the model equations of phytoplankton carbon biomass.

Thank you for bringing this to our attention. We agree and have removed reference to the mean state analysis. We have removed the mean state MLR analysis subplots in Figures S5a and S6a (now labeled S7 and S3 respectively), as well as the regional mean state analysis shown in Figure S7. This helps draw the reader's attention to the changes in internal variance, the main focus of our paper.

[Figure]

"Figure S7: Reconstructed global changes in the contribution of each variable to changes in phytoplankton biomass variance across the RCP8.5 forcing scenario (2006 to 2100). The beginning of the century is shown in light blue and the end of the century is shown in dark blue. The change in variance is calculated using averages across the first (2006 to 2016) and last (2090 to 2100) decades of the RCP8.5 forcing scenario."

[Figure]

"Figure S3: Reconstructed contributions of each variable to phytoplankton biomass variance at the beginning of the RCP8.5 forcing scenario (2006 to 2016). The contribution of cross terms to the MLR reconstruction is shown in the rightmost bar. The variance is calculated using the average across the first (2006 to 2016) decade of the RCP8.5 forcing scenario."

Specific comments:

All line numbers refer to those in the revised manuscript.

L29-: Any reference? And, does this mean the CESM1 shows the opposite response of the high-latitude biomass to the global warming? (Figure 3a shows increase in biomass only in the sea-ice biome).

Thank you for this suggestion. We have included the following citations in the text for clarity.

Steinacher, M., Joos, F., Frölicher, T., Bopp, L., Cadule, P., Cocco, V., Doney, S., Gehlen, M., Lindsay, K., Moore, J., Schneider, B., 535 and Segschneider, J.: Projected 21st century decrease in marine productivity: A multi-model analysis, Biogeosciences, 7, 979–1005, https://doi.org/10.5194/bg-7-979-2010, 2010.

Bopp, L., Resplandy, L., Orr, J., Doney, S., Dunne, J., Gehlen, M., Halloran, P., Heinze, C., Ilyina, T., Séférian, R., Tjiputra, J., and Vichi, M.: Multiple stressors of ocean ecosystems in the 21st century: Projections with CMIP5 models, Biogeosciences, 10, 6225–6245, 350 https://doi.org/10.5194/bg-10-6225-2013, 2013.

Line 28: "A majority of Earth System Models (ESMs) project an increase in phytoplankton abundance in the high latitude ocean as light limitation is alleviated by stratification, increasing temperature stimulates photosynthesis, and sea ice cover declines (Steinacher et al., 2010; Bopp et al., 2013)."

The response of the CESM1-LE is consistent with other CMIP5 simulations which show increasing phytoplankton biomass in the high latitude regions of the Arctic and Southern Oceans with warming. However, the exact distribution of increasing phytoplankton biomass varies between different models. In some models, the increase in phytoplankton biomass is confined to

the sea ice biomes (as seen in the CESM1-LE), while in others there is a broader distribution throughout parts of the Northern Atlantic and Southern Ocean.

L49-: Please elaborate "Clarifying how variance in phytoplankton biomass may be changing over long time scales with climate change is important for fisheries management, especially at regional scales." What kind of impacts on fisheries by changing in variance in Phytoplankton biomass can one expect?

Understanding how variance in phytoplankton biomass is changing in a warming climate is important because it affects our ability to make accurate near-term predictions of fisheries production. We have clarified this point in the text.

Line 48: "… Clarifying how variance in phytoplankton biomass may be changing over long time scales with climate change is important for fisheries management, especially at regional scales, as it affects our ability to make accurate near-term predictions of fisheries production."

L82-85: I could not understand clearly. Please clarify with showing equations.

This is an excellent point. The existing Equation (5) in the Supplemental Information describes the parameterization of zooplankton grazing rate in the CESM1-LE. We have referenced this Equation (5) in the text to clarify.

Line 82: "Grazing rate is computed using a Holling Type III (sigmoidal) relationship and is a function of both prey density and temperature (Figure S1, Equation 5)."

L94-97: The author's description of the experimental setting of CESM1 large ensemble is inaccurate. Please describe it correctly.

We have changed the text to read:

Line 95: "The CESM1-LE simulates the evolution of the climate system from 1920 to 2100 with multiple ensemble members, each expressing different phasing of internal climate variability while responding to a shared external forcing prescription (Kay et al. 2015)."

L99-101: Show figure as an example.

We have removed this sentence for clarity.

L118-120: Please provide the map of the aggregated biological provinces used in this study as supplementally figure or superpose the biome boundary on the main figures (e.g., Figure 3).

This is a great suggestion! We have included a map of the aggregated biomes as cited in Tagliabue et al., 2021 in the Supplemental Information as Figure S2. We have also referenced this map in the text.

[Figure]

"Figure S2: The 11 ecological provinces defined in Tagliabue et al., (2021) and Vichi et al., (2011). Provinces were aggregated using multivariate statistical analysis of physical (i.e., salinity, temperature, mixed layer depth) and biological (i.e., chlorophyll concentration) ocean parameters to group ocean regions with similar physical and environmental conditions. Figure adapted from Tagliabue et al., (2021)."

Line 113: "We classified the marine environmental into 11 ecologically cohesive biomes as in Tagliabue et al., (2021) and Vichi et al., (2011) (Figure S2), which are a consolidation of the 38 ecological regions defined in Longhurst et al., (2007)."

Figure 1: Please use the same colormap and same value range for fair comparison. And, it is better to show the ensemble mean of the σtemporal with a rank analysis (to show whether the observational σ is inside the ensemble spread grid by grid).

Thank you for this suggestion. We have revised Figure 1 to display both maps on the same colormap. However, when the colormaps are the same range it is not possible to see spatial nuances in the remotely sensed plot (Figure 1a). We now note for the reader that we are using different magnitude colorbars in the figure caption, and we added text to reference Table S1 showing the temporal standard deviation differences between model and observation in each province.

[Figure]

"Figure 1: Comparison between observed and modeled phytoplankton biomass interannual variability. (a) Temporal standard deviation in annual mean phytoplankton carbon concentration (mg m$^{-3}$) reconstructed from remotely sensed chlorophyll concentrations, backscattering coefficients, and phytoplankton absorption (1998 to 2019) (Bellacicco et al., 2020). (b) Temporal

standard deviation in annual mean phytoplankton carbon concentration (mg m$^{-3}$) simulated by ensemble member 1 of the CESM1-LE over the same observational period (1998 to 2019). Note the different magnitudes on the colorbars."

| Biome | Name | $\sigma_{temporal, model}$ | $\sigma_{temporal, obs}$ |
|-------|------|---------------------------|--------------------------|
| ARC | Arctic | 2.7 | 4.5 |
| ASP | Atlantic subpolar | 9.7 | 4.1 |
| NAS | North Atlantic subtropical gyre | 2.8 | 1.7 |
| EQA | Equatorial Atlantic | 1.3 | 1.4 |
| SAS | South Atlantic subtropical gyre | 1.1 | 1.2 |
| IND | Indian Ocean | 0.81 | 2.0 |
| SAP | subarctic Pacific | 3.7 | 4.0 |
| NPS | North Pacific subtropical gyre | 0.85 | 1.5 |
| EQP | Equatorial Pacific | 5.8 | 1.8 |
| SPS | South Pacific subtropical gyre | 0.60 | 0.93 |
| SOC | Southern Ocean | 2.7 | 2.7 |

Table S1: " The temporal standard deviation of phytoplankton biomass ($\sigma_{temporal}$) for ensemble member 1 of the CESM1-LE and the satellite observations (Bellacicco et al., 2020) from 1998 to 2019 averaged across the 11 ecological provinces defined in Vichi et al. (2011) and Tagliabue et al. (2021). Units are mg m$^{-3}$."

Line 163: "Some regions of the global ocean display a substantial mismatch in temporal variability between the model and that estimated from observations (Figure 1, Table S1). While the differences can be quite large in some regions, we note that this is an evaluation of temporal variability (rather than internal variability, the focus of this study), and that estimates from the satellite product derive from a collection of data products which may also display biases (Table S1)."

Line 176: "Taken together, our model validation exercises demonstrate that the model tends to overestimate both the temporal (interannual) variability and the internal variability in phytoplankton as compared to satellite observations on both global and regional scales. Thus, we must interpret our findings with this caveat in mind. The change in variance that we model is likely an upper-end estimate."

Line 287: "While the CESM1-LE represents the overall spatial pattern of observed interannual variability in phytoplankton carbon, the model overestimates the magnitude of observed interannual and internal variance in phytoplankton on regional scales…"

L179: Figure 2d?

We believe this to be a mistake on the part of the reviewer. We reference Figure 2b (not 2d) to illustrate the changing coefficient of variance in biomass carbon concentration. There is no Figure 2d.

L213-216: Which regions did the author chose? Please show these on map.

We have included a map of the aggregated biomes as cited in Tagliabue et al., 2021 in the Supplemental Information as Figure S2. We have also referenced this map in the text.

[Figure]

"Figure S2: The 11 ecological provinces defined in Tagliabue et al., (2021) and Vichi et al., (2011). Provinces were aggregated using multivariate statistical analysis of physical (i.e., salinity, temperature, mixed layer depth) and biological (i.e., chlorophyll concentration) ocean parameters to group ocean regions with similar physical and environmental conditions. Figure adapted from Tagliabue et al., (2021)."

Line 113: "We classified the marine environmental into 11 ecologically cohesive biomes as in Tagliabue et al., (2021) and Vichi et al., (2011) (Figure S2), which are a consolidation of the 38 ecological regions defined in Longhurst et al., (2007)."

Technical corrections:

I don't list any small technical/editorial corrections at this time. Above-mentioned conceptual/major comments should be addressed or fixed by the authors before going into details.

---

## Author Comment (AC2)

Review 2 of Elsworth et al. (2022) "Anthropogenic climate change drives non-stationary phytoplankton variance", submitted to Biogeosciences.

The manuscript "Anthropogenic climate change drives non-stationary phytoplankton variance", summarizes projected changes in global and regional phytoplankton variability using the NCAR CESM1 Large Ensemble under a high emissions scenario. The authors explore the key drivers of declining phytoplankton variability, highlighting the importance of top-down, zooplankton grazing in potentially driving future phytoplankton response.

Generally, the article concisely represents its findings but there are several points of clarification I would recommend. In particular, the use of specific statistical terminology could be more accurate. Multiple times throughout the text, the term "variance" is used when, I think, "variability" is intended. In many cases this "variability" is being assessed via the standard deviation of the large ensemble members which is similar to the variance by not the same. Additionally, I am not proficient in MLR, but the comments made in the prior review are troubling especially considering the results are key to the paper's conclusions regarding top-down controls but these results seem underrepresented in the primary manuscript text. I've included several additional minor comments and suggestions below pertaining to clarity and organization.

We have clarified the use of "variance" and "standard deviation" in the text.

We have addressed Reviewer 1's comments regarding the MLR approach in our response above.

Specific Comments and Suggestions:

**Lines 49-52:** *Clarifying how variance in phytoplankton biomass may be changing over long time scales with climate change is important for fisheries management, especially at regional scales. Near- term predictions of phytoplankton biomass may also benefit from knowledge of the projected magnitude of internal variability, as the chaotic nature of internal variability hampers near-term predictions (Meehl et al., 2009, 2014).*

I think it's worth noting that the internal variability quantified using a large ensemble is Internal variability specific to the model and indicative of our uncertainty that results from its simplified representations of the real world processes and numerics. It doesn't necessarily have any bearing on real world manifestations of variability. Its primary utility to management and fisheries is in guiding our level of confidence in disentangling model signals from the noise.

This is an excellent point. We have included additional text to clarify this point.

Line 53: "… However, modeled internal variability may differ from that observed in the real world."

**Lines 103-104:** *Six CESM1-LE members had corrupted ocean biogeochemistry*

I'm curious, what does "corrupted ocean biochemistry" mean? it might help to explain what makes an ensemble member usable versus not.

The ocean biogeochemistry output fields of ensemble members 3 though 8 were corrupted during the saving process. Therefore, no information on biogeochemical variables is available for these ensemble members. However, the corruption of these ensembles affected only the biogeochemical output and, thus, other Earth system variables is preserved. Details are referenced here.

**Figure 1.** Add units: standard deviation should have the same units as the variable being assessed (i.e., phytoplankton carbon) but none appear in figure 1.

Thank you for bringing this to our attention. We have revised Figure 1 to include the units of mg m$^{-3}$ and have modified the text of the figure caption.

[Figure]

"Figure 1: Comparison between observed and modeled phytoplankton biomass interannual variability. (a) Temporal standard deviation in annual mean phytoplankton carbon concentration (mg m$^{-3}$) reconstructed from remotely sensed chlorophyll concentrations, backscattering coefficients, and phytoplankton absorption (1998 to 2019) (Bellacicco et al., 2020). (b) Temporal standard deviation in annual mean phytoplankton carbon concentration (mg m$^{-3}$) simulated by ensemble member 1 of the CESM1-LE over the same observational period (1998 to 2019). Note the different magnitudes on the colorbars."

**Lines 121-122:** *Internal variability at each location (x,y) is approximated as the standard deviation across ensemble members (EMs) at a given time (t)*

The method described here indicates that the standard deviation is being used to quantify variability. However, throughout the paper, the authors reference the "variance" when I think they mean "variability". This is problematic because "variance" and "standard deviation", while related, are two different values and the way they are interchanged throughout the text is confusing. Please check all instances of "variance" in the paper for intended meaning and replace with "variability" where appropriate. I suggest including a description of the "coefficient of variance" method here, too.

We agree. We have included a description of the coefficient of variance in the methods section.

Line 130: "The coefficient of variance (CoV) is calculated as the standard deviation across the ensemble members divided by the ensemble mean,

$$CoV\ (x,\ y,\ t) = \frac{\sigma(EM\ (x,\ y,\ t))}{LE}$$"

**Lines 142-143:** *However, while the model ensemble captures regional patterns of observed variability, the CESM1-LE overestimates the magnitude of observed interannual variability.*

I may be mistaken but it seems this was determined using only a single ensemble member - is it appropriate for conclusions to be drawn for the full ensemble when only considering one ensemble member?

We tested the temporal standard deviation for all ensemble members and report the (small) difference for the reviewer. The figure below shows the temporal standard deviation of 34 ensemble members of the CESM1-LE across the observational period (1998 to 2019). The difference in temporal standard deviation between ensemble members is small over this period.

[Figure]

Figure: Histogram of global average temporal standard deviation (mg m$^{-3}$) for each of the 34 ensemble members of the CESM1-LE over the observational window (1998 to 2019).

**Lines 147:** *A synthetic ensemble is a novel technique*

I don't think this technique can be called "novel" if it appears in two prior references

This is a valid point. We have removed the word "novel" from the text on Line 147 and Line 590.

**Lines 149-151:** *Compared to the internal variability over the observational period (2002 to 2020) (purple circle, (Figure S2), the model ensemble slightly overestimates the magnitude of internal variability in chlorophyll observed in the real world.*

This seems like a result/ should appear in the result section. Also, it makes an assessment of the ensemble as a whole, but isn't it still based on the results from the single ensemble member? If not, this was a point of confusion for me, and I suggest clarifying.

This is our second model validation exercise, and thus we opted to keep it in the methods section. We note for the reviewer that this is an assessment of internal variability (ensemble spread), as compared to a synthetic ensemble generated from observations. In response to this comment, we added a paragraph describing the interpretation of the results from the interannual and internal variance validation exercises to the methods section.

Line 163: "Some regions of the global ocean display a substantial mismatch in temporal variability between the model and that estimated from observations (Figure 1, Table S1). While the differences can be quite large in some regions, we note that this is an evaluation of temporal variability (rather than internal variability, the focus of this study), and that estimates from the satellite product derive from a collection of data products which may also display biases (Table S1)."

Line 176: "Taken together, our model validation exercises demonstrate that the model tends to overestimate both the temporal (interannual) variability and the internal variability in phytoplankton as compared to satellite observations on both global and regional scales. Thus, we must interpret our findings with this caveat in mind. The change in variance that we model is likely an upper-end estimate."

**Lines 153-154:** *Annually averaged, global mean, upper-ocean (top 150m) integrated phytoplankton biomass across the model ensemble decreases from 76.1 mmol C m-2 to 66.2 mmol C m-2.*

It's not clear what timeframes these values represent. Is it 2006 vs. 2100? If so, it seems that such a narrow, 1-year window would risk aliasing higher frequency variability and potentially under- or overestimate the change in mean state. This is somewhat compensated for by the size of the ensemble but differs from the 10-year averaging described later in Line 223

This is an excellent point. The decline in phytoplankton biomass is calculated as the difference between the average of the first (1920 to 1930) and last (2090 to 2100) decades across the historical and the RCP8.5 forcing scenario. We have clarified the time windows used in this calculation in the manuscript text.

Line 181: "The change in the mean is calculated as the difference between the first (1920 to 1930) and last (2090 to 2100) decades across the historical and the RCP8.5 forcing scenario."

**Lines 177-178:** *we calculated the coefficient of variance as the standard deviation across the ensemble members for a given year (ensemble spread) divided by the ensemble mean.*

I suggest including this description in the methods section rather than the results.

Thank you for this suggestion. We have included a description of the coefficient of variance in the methods section.

Line 130: "The coefficient of variance (CoV) is calculated as the standard deviation across the ensemble members divided by the ensemble mean,

CoV $(x, y, t) = \dfrac{\sigma(EM\ (x, y, t))}{LE}$"

**Lines 178-180:** *Figure 2b illustrates the change in the coefficient of variance from the historical period through the RCP8.5 forcing scenario (1920 to 2100).*

The results seem to jump from Figure 2a, to Figure 3, then back to 2b which is a bit confusing.

This is an excellent point. We have modified the text to enhance the flow of the manuscript.

**Line 180:** *The coefficient of variance is relatively constant across the historical period (1920 to 2005), and then significantly declines by ~20% from 2006-2100.*

I'm not sure I agree with the assessment that the coefficient of variance is relatively constant across the historical period. 1920-1980 appears to have a positive trend with a range of about 6.1 to 7.3, which appears similar to the range of the time period covered by the dashed line in Figure 2b. I suggest testing the significance of the 1920-1980 trend. Also, could the drop in coefficient of variance instead be explained by temporal distance from the perturbation that differentiates the ensemble members? If the 34 ensemble members differ in initial air temperature conditions, would the spread perhaps be expected to decrease as the simulation integrates further away from that initial discrepancy (i.e., solutions start to converge)?

This is a good point. We have tested the significance of the 1920 to 1980 trend and find that it is not significant.

However, the decrease in the coefficient of variance over the course of the simulation is not due to an increase in the time since the ensemble members were perturbed. This has been demonstrated in the study "An Ensemble Covariance Framework for Quantifying Forced Climate Variability and Its Time of Emergence" published by Yettella et al., 2018. This is illustrated by different responses of ocean and land variance over the 21$^{st}$ century, with ocean variance declining and land variance increasing over time.

**Lines 190-193:** *From 2006 to 2100, the coefficient of variance decreases by 3.3 x 10-5 yr-1 in the CESM1-LE, 2.0x10-4 yr in the MPI-ESM-LR1, 5.2x10-5 yr-1 in the CanESM2, and 3.9 x10-4 yr-1 in the GFDL-ESM2M. These declines are statistically significant in all model ensembles with the exception of the MPI-ESM-LR1 (Figure S2).*

It's not clear how these values across models are calculated, whether the end points of the time series or a range of years - the latter would be more appropriate (as done in Line 223) to avoid higher frequency variability and thus under- or overestimating the nature of the change. I also

suggest reporting the specific statistical testing methods in the text if stating that the changes are significant.

Thank you for mentioning this. Changes in the coefficient of variance are calculated using averages of the first (2006 to 2016) and last (2090 to 2100) decades of the RCP8.5 forcing scenario. We have clarified this in the caption of Figure S2 (now Figure S4) and in the manuscript text. Significance of the trends are determined by a t-test with a p-value less than 0.05. We have also included this in the caption of Figure S2 (now Figure S4).

Line 220: "… The change in the coefficient of variance is calculated using averages across the first (2006 to 2016) and last (2090 to 2100) decades of the RCP8.5 forcing scenario."

Figure S2 (now S4): "… Trend significance is determined by a *t*-test with a p-value less than 0.05."

**Line 201:** *We observe the largest magnitude decline in total phytoplankton carbon variance*

The table is reporting change in standard deviation, not variance. Standard deviation is expressed in the same units as the analyzed variable while variance is reported in the square of those units.

Thank you for clarifying. We have changed "variance" to "standard deviation" in the text to be more precise.

Line 229: "Global changes in total phytoplankton biomass standard deviation are a manifestation of changes in diatom and small phytoplankton variability (Table 1). We observe the largest magnitude decline in total phytoplankton carbon standard deviation in the subpolar Atlantic (ASP) region, where diatom standard deviation declines by ~10 mmol C m$^{-2}$ and small phytoplankton standard deviation declines by ~2 mmol C m$^{-2}$ (Table 1). The CESM1-LE simulates a moderate magnitude decline in total phytoplankton standard deviation in the subarctic Pacific (SAP) region, driven by a decrease in small phytoplankton standard deviation (~2 mmol C m$^{-2}$) with minor contributions from declines in diatom standard deviation (~1 mmol C m$^{-2}$) (Table 1). Moderate declines in standard deviation are also simulated in the Arctic (ARC), North Atlantic subtropical gyre (NAS), Southern Ocean (SOC), and Equatorial Pacific (EQP) regions, driven by declines in diatom carbon standard deviation in the SOC region and declines in small phytoplankton variance in the EQP region (Table 1)."

Figure 4: It's not clear what this figure adds to the discussion - it seems to be redundant with information in Figure 5. Perhaps if the outlines of the ecological regions were included?

This is an excellent suggestion. We have included a map of the aggregated biomes as cited in Tagliabue et al., 2021 in the Supplemental Information as Figure S2. We have also referenced this map in the text.

[Figure]

"Figure S2: The 11 ecological provinces defined in Tagliabue et al., (2021) and Vichi et al., (2011). Provinces were aggregated using multivariate statistical analysis of physical (i.e., salinity, temperature, mixed layer depth) and biological (i.e., chlorophyll concentration) ocean parameters to group ocean regions with similar physical and environmental conditions. Figure adapted from Tagliabue et al., (2021)."

Line 113: "We classified the marine environmental into 11 ecologically cohesive biomes as in Tagliabue et al., (2021) and Vichi et al., (2011) (Figure S2), which are a consolidation of the 38 ecological regions defined in Longhurst et al., (2007)."

**Lines 219-221:** *We quantified the relationship between phytoplankton carbon and the variables which contribute to changing phytoplankton biomass and its internal variability by performing a multiple linear regression (MLR) analysis. The MLR analysis was performed on linearly detrended annual anomalies using the ordinary least squares function of the Python package statsmodels.api*

This and the associated equations seem to belong in the methods section.

Thank you for this suggestion. We have moved this and the associated equations to the methods section.

Line 134: "… We quantified the relationship between phytoplankton carbon and the variables which contribute to changing phytoplankton biomass and its internal variability by performing a multiple linear regression (MLR) analysis. The MLR analysis was performed on linearly detrended annual anomalies using the ordinary least squares function of the Python package statsmodel.api."

**Line 274:** *…and important global biogeochemical regions…*

What is considered an important biogeochemical region? This seems somewhat vague - I suggest elaborating to be a bit more specific.

We appreciate this comment. An important biogeochemical region is an ocean region characterized by coherent physical and environmental conditions, which support unique marine ecosystems with an outsized role on ocean biogeochemical cycling. We have added a sentence in the text to clarify this point.

Line 123: "… Important biogeochemical regions are those characterized by coherent physical and environmental conditions, which support unique marine ecosystems with an outsized role on global ocean biogeochemistry."

**Lines 278-280:** *As such, the magnitude of changes in phytoplankton internal variance derived from the model ensemble should be interpreted as an overestimate when considering changes in phytoplankton internal variance driven by anthropogenic warming.*

Again, my impression was that this conclusion was derived from analyzing a single ensemble member which seems insufficient for assessing the entire ensemble.

This conclusion was derived from our second model validation exercise, where we compare the spread across all modeled ensemble members with that of a synthetic ensemble derived from satellite observations.

---

## Author Comment (AC3)

Review 3 of Elsworth et al. (2022) "Anthropogenic climate change drives non-stationary phytoplankton variance", submitted to Biogeosciences.

In this manuscript, the authors use the Community Earth System Model I Large Ensemble to evaluate the impacts of anthropogenic climate change on long-term variability in phytoplankton distributions within the global ocean. The authors additionally use a multiple linear regression to evaluate the ecological drivers of this change, reporting zooplankton grazing as being a major factor in reducing variability in phytoplankton biomass.

The analysis of earth systems models is well outside my area of expertise. So while the authors' main finding that variance in phytoplankton biomass is anticipated to decrease in the future ocean seems informative from my perspective, I defer to the first reviewer's comments regarding best practices in model interpretation. I was interested to see the multiple linear regression results, which seem to highlight a particularly strong coupling between phytoplankton biomass and grazing in model results. However, by the authors' admission on L265, it does not seem possible to establish cause and effect regarding the nature of this interaction. With this, it seems like an overstatement to suggest (as in the abstract and elsewhere) that these results provide evidence for grazing-driven declines in phytoplankton biomass.

More importantly, insufficient documentation is provided for the reader to interpret the MLR results. Critically, it is not immediately clear from the text how contributions to phytoplankton/diatom variance were calculated. Equations should be provided, and associated details on the MLR analysis should be moved to the methods section to make this information easier to locate in the manuscript. Moreover, the MLR results themselves seem insufficiently documented. No details are provided on the overall model fit nor on uncertainties associated with the MLR coefficients. The relationship between the parameters   quationns 3 and 4 and the larger set of parameters included in figure 5 is unclear as well.

The discussion should also be expanded to provide more context on the authors' interpretation of these results. Altogether, even after reading the manuscript several times, I'm not sure why the results shouldn't be interpreted as a weakening of top-down control in the future ocean (with the decrease in contributions to phytoplankton biomass variance due to grazing in Figure 5 reflecting a reduced coupling of phytoplankton biomass and grazing and, by extension, a strengthening of bottom-up controls). If this interpretation is beyond what can be determined based on the analysis (for instance because of large uncertainties in coefficient errors), this is not evident from the information provided.

Without this information on the MLR results, it is impossible to critically evaluate some of the the manuscript's main conclusions. With this, and in light of the comments made by the first reviewer regarding issues with the authors' analysis of the CESM results, I cannot recommend this manuscript for publication without major revisions. A few specific comments are provided below.

We agree with the reviewer. We have made multiple changes to the wording in the manuscript to address this point. These modifications are listed below.

We have changed the text on Line XX: "In these high-latitude regions, bottom-up controls (e.g., light, nutrients) have only a small effect on biomass variance. Rather, the declining internal variance in phytoplankton biomass co-occurs with a reduction in zooplankton grazing variability. Similar patterns emerge in the biogeochemically critical regions of the Southern Ocean and the Equatorial Pacific."

We have changed all wording about "driver / drivers" to "contribution / contributions" throughout the manuscript.

We have modified the topic sentence of the paragraph Line XX: "The declining internal variance in phytoplankton biomass co-occurs with a reduction in zooplankton grazing variability".

We have changed the language on Line XX: "The declining internal variance in phytoplankton biomass co-occurs with a reduction in zooplankton grazing variability." And on Line 275: "…".

We have modified the text on Line XX: "Statistical analysis of these specific regions reveal the decline in phytoplankton biomass variance to co-occur with a reduction in zooplankton grazing variability, consistent with previous studies (Bopp et al., 2001; Laufkötter et al., 2015)."

We have changed Line 289: "our study demonstrates a strong connectivity between phytoplankton and zooplankton grazing variance…."

We have changed Line XX: "Our study demonstrates a strong connectivity between phytoplankton and zooplankton grazing variance in the subpolar North Atlantic and the subpolar North Pacific."

We have modified Line 307: "Our regional analyses suggest that both phytoplankton and zooplankton grazing variance are likely to change with anthropogenic warming."

We have clarified Line XX: "However, our regional analyses suggest that both phytoplankton and zooplankton grazing variance are likely to change with anthropogenic warming."

**Specific comments:**

L114 – 115 – A quick review of the method used in Tagliabue et al. would be useful here. What were the multivariate statistical methods used? How were they applied? A map of the biomes would be informative as well.

This is an excellent suggestion. We've included a description of the methods used by Vichi et al. 2011.

Line 113: "We classified the marine environment into 11 ecological cohesive biomes as in Tagliabue et al., 2021 and Vichi et al., 2011, which are a consolidation of the 38 ecological regions defined in Longhurst et al., 2007. The provinces were aggregated using multivariate statistical analysis of physical (i.e., salinity, temperature, mixed layer depth) and biological (i.e., chlorophyll concentration) ocean parameters to group ocean regions with similar physical and environmental conditions (Vichi et al., 2011). Analyses were performed by randomly selecting from a

combination of model and observational datasets and testing for statistical significance using analysis of similarities (ANOSIM) (Vichi et al., 2011)."

L159 – 161 - This text feels more appropriate for conclusion/discussion.

We agree. We have removed this text from the manuscript.

"In the North Atlantic subpolar gyre, the phytoplankton biomass declines by 40-50% of its mean (Figures 3a, S3a)."

L215 - FAO citation and the associated reference seem to be improperly formatted

Thank you for this comment. We have reformatted the FAO citation in the references. The references is cited parenthetically as (*FAO, 2020*). We will ask the editorial staff for clarification if/when the manuscript is accepted.

"FAO. 2020. *The State of World Fisheries and Aquaculture 2020. Sustainability in action.* Rome."

L219 – 234 - This text feels more appropriate for the methods section

Thank you for this suggestion. We have moved this text to the methods section.

L289 – 291 - Is this conclusion inconsistent with the disclaimer provided at L264 – 266?

Equations 3 & 4 — Why are the terms in the equations (e.g., Solar, SST, Nutrient, etc.), different from those included in figure 5? Were the equations in the text just providing a summary of the actual equations used? If so, this should be made explicitly clear, with some description of all the variables included.

Yes, the terms in the equations (e.g., Solar, SST, Nutrient, etc.) are the same as those included in Figure 5. We have added the terms parenthetically below the variable names in Figure 5 to clarify this point.

[Figure]

Figure 5: "Reconstructed changes in the contribution of each variable to phytoplankton biomass standard deviation across the RCP8.5 forcing scenario (2006 to 2100) with the beginning of the century shown in light blue and the end of the century shown in dark blue. Marine ecological regions are defined in Tagliabue et al. (2021). Regions were selected which aligned with the highest fisheries catch in the (a) Atlantic and (b) Pacific basins and the biogeochemically important (c) Southern Ocean and (d) Equatorial Pacific regions. The dominant phytoplankton functional type is considered in each region. In regions with a mixed ecological assemblage, total phytoplankton carbon is considered. The change in the coefficient of variance is calculated using averages across the first (2006 to 2016) and last (2090 to 2100) decades of the RCP8.5 forcing scenario."

Figure 4 — Minor tick marks not necessary on color scale; difficult to see regions dominated by diazotrophs. Maybe use color palette with more contrast?

Thank you for this suggestion. We have removed unnecessary tick marks from the color scale. Diazotrophs do not dominate in any regions of the global ocean and are not visible on Figure 4 for this reason.

[Figure]

"Figure 4: Distribution of the dominant phytoplankton functional type in biomass carbon averaged across the RCP8.5 forcing scenario (2006 to 2100). The CESM1-LE simulates three phytoplankton functional types: diatoms, diazotrophs, and small phytoplankton. Regions where diatoms dominate

are shown in yellow, regions where diazotrophs dominate are shown in pink, and regions where small phytoplankton dominate are shown in purple."

Figure 5 —Note inconsistent capitalization of biomes in subplots; Are units correctly labeled? Are the units for "contribution to phytoplankton/diatom variance" really mmol C m$^{-2}$? On a related note, where did the values on the Y axis come from? Based on the axis label they don't correspond to the MLR coefficients, but I didn't see any details in the text.

Thank you for clarifying. We have changed the text in the figure caption to clarify that we show the phytoplankton biomass standard deviation.

We follow the province labels set forth in Tagliabue et al., 2021 in both Table 1 and Figure 5. Proper nouns are capitalized (e.g., Equatorial Pacific, Southern Ocean) while adjectives are lowercase (e.g., Atlantic subpolar, South Pacific subtropical gyre).

[Figure]

Figure 5: "Reconstructed changes in the contribution of each variable to phytoplankton biomass standard deviation across the RCP8.5 forcing scenario (2006 to 2100) with the beginning of the century shown in light blue and the end of the century shown in dark blue. Marine ecological regions are defined in Tagliabue et al. (2021). Regions were selected which aligned with the highest fisheries catch in the (a) Atlantic and (b) Pacific basins and the biogeochemically important (c) Southern Ocean and (d) Equatorial Pacific regions. The dominant phytoplankton functional type is considered in each region. In regions with a mixed ecological assemblage, total phytoplankton carbon is considered. The change in the coefficient of variance is calculated using averages across the first (2006 to 2016) and last (2090 to 2100) decades of the RCP8.5 forcing scenario."

---

## Referee Report (RR1)

**Review report on "Anthropogenic climate change drives non-stationary phytoplankton variance"**

This is a very interesting study on the impacts of climate change on the internal variability of phytoplankton, based on an Earth System Model ensemble. While I believe this study is quite important and of interest in terms of results (the main result being a reduction in phytoplankton internal variability under anthropogenic driven climate change) as in terms of implications (especially the link with fisheries stock assessment, whose uncertainty could be reduced as a consequence of this reduced phytoplankton internal variability), I have major concerns about the MLR method used to estimate the physical and biogeochemical drivers of trends in phytoplankton internal variability. I elaborate on these and other issues below.

**MAJOR COMMENTS:**

**My first concern is the choice of explanatory variables for the MLR:**

MLR is a great tool for exploring relationships between variables, but as you have indicated in the text, it is unable to distinguish between bottom-up and top-down relationships that link two variables. For these reasons, in order to be able to interpret the results with causality relationship, you should :

- 1) Use only variables for which the causal relationship with phytoplankton biomass is known (or for which the first order of this relationship is known), e.g., SST (the first order is a bottom-up relationship: warming drives phytoplankton biomass by increasing metabolic rates, a positive effect, and by increasing nutrient stratification, a negative effect. At second order, one could have a top-down feedback of phytoplankton biomass change modifying carbon cycling and indirectly temperature, but one would neglect this effect). For this reason, I think that including zooplankton/zooplankton grazing in such an analysis is not appropriate because you are not able to separate top-down and bottom-up effects on phytoplankton biomass.
- 2) Use only variables for which the causal relationship with the target variable is the same. In your case, use the variables for which phytoplankton biomass is a consequence, not a cause. Again, while I believe that zooplankton do exert top-down control over phytoplankton (so that phytoplankton biomass would be a consequence), PFT models with small numbers of zooplankton are likely to be dominated by bottom-up control (so that zooplankton would be the consequence and phytoplankton the cause). To support this claim, trophic amplification under climate change in these models has been described as a good indicator of bottom-up control of zooplankton by phytoplankton (Chust et al., 2014, Kwiatkowski et al., 2019), a pattern that is altered when higher trophic levels are considered (Dupont et al., 2022).
- 3) MLR analysis assumes independence of explanatory variables, which is clearly not the case (e.g., MLD and Nutrient are highly correlated). I agree that this is a classic problem in multivariate analysis on climate variables, but this point should be discussed further, by providing at least one correlation matrix between all explanatory variables.

Nevertheless, your signal on zooplankton is clearly related to the strong relationship between zooplankton and phytoplankton, which is expected but clearly interesting. I think you should analyze (with simple linear regression) the relationship between phytoplankton and zooplankton separately from the other variables, which would clearly fit the main message of your paper: showing that the effect of climate change on the internal variability of phytoplankton is transferred to the internal variability of zooplankton would demonstrate a transfer of the trend in internal variability to the higher trophic levels (in this case, zooplankton), which you could then extrapolate in the discussion to even higher trophic levels (e.g. fish). It would also be interesting to compare trends in phytoplankton and zooplankton internal variability. Is it higher? Lower? Why? You could also do the same MLR analysis with trends in total plankton with the bottom-up effect.

**My second concern is about the MLR method itself, which I think is wrong in its current form:**

First of all, I am missing some details to understand what exactly you did with the MLR. You would need to make it clear which variables are used for each step of the method. In particular, it is not clear which variable depends on i) time, ii) space, and iii) the member of the model set. For the rest of my argument, *t* will refer to

time, *x* to the grid cell (spatial position) and *i* to the model set member. *Y* will refer to the phytoplankton biomass and *X* to any explanatory variable.

A) So, for what I understand, your first step was to prepare linearly detrended annual anomalies. So, for a variable *X*, with a trend *a*, the considered variable in the MLR is

**$X^{d}(t,x,i)=(X(t,x,i)-X(0,x,i))-a(x)^{*}t$**

, a field with 3 dimensions : space, time and model ensemble member. The same calculation gives you  $Y^{d}(t,x,i)$ . If it is based on globally averaged values, I would recommend to keep the space dimension.

B) With the MLR, you fitted the following relationship and thus estimated the coefficients  $dY^d/dX^d$ :

 $Y^{d}(t,x,i) = sum \left( (dY^{d}/dX^{d})^{*} X^{d}(t,x,i) \right)$

An approximation of the first order taylor development which would give

 $Y^{d}(t,x,i) = sum ( (dY^{d}/dX^{d})(t,x,i) * X^{d} (t,x,i))$

C) Then, by linearity, you compute (*t* being a 10-year average)

 $Sigma_i(Y^d(t,x,i)) = sum (dY^d/dX^d sigma_(X^d(t,x,i)))$

BUT : Variance isn't linear (neither is the standard deviation). Even if two variables are independent (which is definitely not the case),  $VAR(aX+bY) = a^2 Var(X) + b^2 Var(Y)$ .

In your case, if you wanted to reconstruct your variance, you would use the following formula ( $a_i$  being your MLR coefficient  $dY^d/dX^d$  and Cov being the covariance and not the coefficient of variance here):

$$egin{aligned} \operatorname{Var}\!\left(\sum_{i=1}^N a_i X_i
ight) &= \sum_{i,j=1}^N a_i a_j \operatorname{Cov}(X_i,X_j) \ &= \sum_{i=1}^N a_i^2 \operatorname{Var}(X_i) + \sum_{i
eq j} a_i a_j \operatorname{Cov}(X_i,X_j) \ &= \sum_{i=1}^N a_i^2 \operatorname{Var}(X_i) + 2\sum_{1\leq i < j \leq N} a_i a_j \operatorname{Cov}(X_i,X_j) \end{aligned}$$

You can calculate this value perfectly well, but I'm not sure that's what you want to do.

Indeed, I'm not sure what kind of information you expect from the relationships between linearly detrended variables and linearly detrended phytoplankton carbon biomass: do you want to explain the internal phytoplankton variability by the internal variability of others variables or do you want to explain it by the trends of other variables ? I think the second option, or both, would be more appropriate (e.g., is the increase in temperature related to the reduction in internal variability of phytoplankton biomass?)

While I think your current method is wrong, I keep in mind that too few details have been provided to be certain, and perhaps I will be convinced of your method when more details are included. Nevertheless, I suggest another approach:

You have introduced CoV (Coefficient of Variance), and I think this variable is indeed more appropriate than standard deviation because it removes the effect of reduced mean state values on the change in internal variability (i.e., a lower mean state will lead to lower internal variability in absolute magnitude, but not necessarily to a reduced coefficient of variance).

You could perform the MLR on CoV directly (which would mean using the mean anomalies of the entire model ensemble), i.e., estimate a linear relationship between  $CoV_phyto(t,x)$  and other variables:

 $CoV_phyto(t,x) = CoV_phyto(0,x) + a * anomaly_var1(t,x) + b * anomaly_var2(t,x)...$

Or if you want-to keep detrended variables:

 $CoV_phyto(t,x) = CoV_phyto(0,x) + a CoV_var1(t,x) + b CoV_var2(t,x)...$

**MINOR COMMENTS :**

**Discussion :** The discussion is quite short, I would like to see a discussion of the mechanisms that might lead to this reduction in internal phytoplankton variance. In addition, the discussion focuses on the top-down control of zooplankton on phytoplankton. Although the authors no longer assert in the current version that zooplankton are a driver of trends in internal phytoplankton variability, they continue to discuss it, which is not necessarily relevant. Given my main comment on how to study the relationship between phytoplankton and zooplankton variability, I would focus on the bottom-up effect of phytoplankton on zooplankton to support the impact of changes in phytoplankton variability on higher trophic levels, and then discuss top-down effects as a limitation to the interpretation of your results.

**Wording :** Consider using "internal variability" instead of "variance" throughout the text, starting with the title. While variance can refer to many temporal scales (seasonal, interannual,...), I think internal variability is much more accurate (e.g., L7, "internal variability" instead of "internal variance")

**Other comments :**

L1: Bopp et al., 2001, 2013; Laufkötter et al., 2015; Kwiatkowski et al., 2020 are model studies. If you want to keep past tense, please add data-based reference. Or use another tense.

L4: I would mention the impact of phytoplankton on the carbon cycle. Also in the discussion.

L45-46: As formulated, the results of the Resplandy's study are not clear.

L54: The last sentence of the paragraph does not flow well with the rest.

Fig 1 and L160: "by ensemble member 1 of the CESM1-LE": why not using the average of the model ensemble members ?

Eq. 1: use a separate symbol for internal variability (you have twice sigma)

Eq. 2: Specify on which variable your mean LE is computed (time, space and model ensemble members)

L132: unclear, why not using the same term as above, i.e., "ensemble mean", i guess that LE(x,y,t) is the same as LE but this needs to be specified.

L134: Please avoid using variance if you refer to internal variability.

L164: use interannual instead of temporal which is no precise

L179 (and L290): I disagree with this statement: the comparison between the observed variance and the modeled variance does not give any information about the trends in the variance.

Fig 3. Specify the variables on which you have applied a t-test, at least as supplementary material. The reader should be able to assess the validity of your test (sample size, normality assumption, etc.).

L239: Why not use the total biomass of phytoplankton everywhere?

Fig 4: Please add the regions to the map (at least ASP, SAP, SOC, and EQP), and consider showing diatom biomass over small phytoplankton biomass ratio, as diazotrophs are not dominant anywhere and to be more quantitative.

L260: The wording of the sentence suggests that zooplankton exert top-down control over phytoplankton, which is uncertain or even false.

L275-279: I think this result (with an appropriate MLR method) is very interesting in explaining what is driving the zooplankton variability (see main comment) and should be interpreted from a zooplankton perspective.

L287: I do not believe that either of these references is relevant to this statement.

**Références:**

Chust, G., Allen, J.I., Bopp, L., Schrum, C., Holt, J., Tsiaras, K., Zavatarelli, M., Chifflet, M., Cannaby, H., Dadou, I., Daewel, U., Wakelin, S.L., Machu, E., Pushpadas, D., Butenschon, M., Artioli, Y., Petihakis, G., Smith, C., Garçon, V., Goubanova, K., Le Vu, B., Fach, B.A., Salihoglu, B., Clementi, E. and Irigoien, X. (2014), Biomass changes and trophic amplification of plankton in a warmer ocean. Glob Change Biol, 20: 2124-2139. https://doi.org/10.1111/gcb.12562

Kwiatkowski, L, Aumont, O, Bopp, L. Consistent trophic amplification of marine biomass declines under climate change. *Glob Change Biol.* 2019; 25: 218– 229. https://doi.org/10.1111/gcb.14468

Dupont, L., Le Mézo, P., Aumont, O., Bopp, L., Clerc, C., Ethé, C., & Maury, O. (2022). High trophic level feedbacks on global ocean carbon uptake and marine ecosystem dynamics under climate change. *Global Change Biology*.

---

## Referee Report (RR2)

Review of ***Anthropogenic climate change drives non-stationary phytoplankton variance*** by Elsworth et al.

This work investigates the internal variability of phytoplankton biomass in an Earth System Model large ensemble (CESM1-LE). The results show a decrease in internal variability under RCP8.5.

The manuscript is well written and well laid out and the results are novel and important. I have only some minor suggestions.

Comments

Line 128: The *coefficient of variance* seems to be much more commonly referred to as the *coefficient of variation*. Is there any reason why you use the term "coefficient of variance"?

Line 130: Should the nominator in this equation read LE(x,y,t) (with a line above) as it is defined on line 132? In that case I would rephrase the sentence on line 131 to say something like: *where LE(x,y,t), the forced response of the large ensemble, is calculated as the mean of ensemble members at a given location and time.*

Line 140: Should be biomass *standard deviation* instead of *variance*? Also on line 145. Also check other places throughout the manuscript like on lines 225-229.

Line 169 - 175: It is not specified in the text that it is *surface* chlorophyll that you are using. Also, could you please clarify why you used surface chlorophyll for the validation of internal variability instead of biomass as in the rest of the analysis?

212:  Specify that it is *surface* chlorophyll.

---

## Author Response (AR2)

Review 1 of Elsworth et al. (2022) "Anthropogenic climate change drives non-stationary phytoplankton variance", submitted to Biogeosciences.

This is a very interesting study on the impacts of climate change on the internal variability of phytoplankton, based on an Earth System Model ensemble. While I believe this study is quite important and of interest in terms of results (the main result being a reduction in phytoplankton internal variability under anthropogenic driven climate change) as in terms of implications(especially the link with fisheries stock assessment, whose uncertainty could be reduced as a consequence of this reduced phytoplankton internal variability), I have major concerns about the MLR method used to estimate the physical and biogeochemical drivers of trends in phytoplankton internal variability. I elaborate on these and other issues below.

**MAJOR COMMENTS:**

My first concern is the choice of explanatory variables for the MLR:

MLR is a great tool for exploring relationships between variables, but as you have indicated in the text, it is unable to distinguish between bottom-up and top-down relationships that link two variables. For these reasons, in order to be able to interpret the results with causality relationship, you should :

- 1. 1) Use only variables for which the causal relationship with phytoplankton biomass is known (or for which the first order of this relationship is known), e.g., SST (the first order is a bottom-up relationship: warming drives phytoplankton biomass by increasing metabolic rates, a positive effect, and by increasing nutrient stratification, a negative effect. At second order, one could have a top- down feedback of phytoplankton biomass change modifying carbon cycling and indirectly temperature, but one would neglect this effect). For this reason, I think that including zooplankton/zooplankton grazing in such an analysis is not appropriate because you are not able to separate top-down and bottom-up effects on phytoplankton biomass.
- 2. 2) Use only variables for which the causal relationship with the target variable is the same. In your case, use the variables for which phytoplankton biomass is a consequence, not a cause. Again, while I believe that zooplankton do exert top-down control over phytoplankton (so that phytoplankton biomass would be a consequence), PFT models with small numbers of zooplankton are likely to be dominated by bottom-up control (so that zooplankton would be the consequence and phytoplankton the cause). To support this claim, trophic amplification under climate change in these models has been described as a good indicator of bottom-up control of zooplankton by phytoplankton (Chust et al., 2014, Kwiatkowski et al., 2019), a pattern that is altered when higher trophic levels are considered (Dupont et al., 2022).
- 3. 3) MLR analysis assumes independence of explanatory variables, which is clearly not the case (e.g., MLD and Nutrient are highly correlated). I agree that this is a classic problem in multivariate analysis on climate variables, but this point should be discussed further, by providing at least one correlation matrix between all explanatory variables.

Nevertheless, your signal on zooplankton is clearly related to the strong relationship between zooplankton and phytoplankton, which is expected but clearly interesting. I think you should analyze (with simple linear regression) the relationship between phytoplankton and zooplankton separately from the other variables, which would clearly fit the main message of your paper: showing that the effect of climate change on the internal variability of phytoplankton is transferred to the internal variability of zooplankton would demonstrate a transfer of the trend in internal variability to the higher trophic levels (in this case, zooplankton), which you could then extrapolate in the discussion to even higher trophic levels (e.g. fish). It would also be interesting to compare trends in phytoplankton and zooplankton internal variability. Is it higher? Lower? Why? You could also do the same MLR analysis with trends in total plankton with the bottom-up effect.

We thank the reviewer for their constructive comments and appreciate their careful reading of the manuscript. We agree that the manuscript would benefit from extending our analysis to include these methods. To more quantitatively assess the relationship between phytoplankton and zooplankton, we will revise the manuscript to include this analytical approach. We thank the reviewer for this correction and will adapt our results section accordingly.

**My second concern is about the MLR method itself, which I think is wrong in its current form:**

First of all, I am missing some details to understand what exactly you did with the MLR. You would need to make it clear which variables are used for each step of the method. In particular, it is not clear which variable depends on i) time, ii) space, and iii) the member of the model set. For the rest of my argument, *t* will refer to

time, x to the grid cell (spatial position) and i to the model set member. Y will refer to the phytoplankton biomass and X to any explanatory variable.

A) So, for what I understand, your first step was to prepare linearly detrended annual anomalies. So, for a variable *X*, with a trend *a*, the considered variable in the MLR is

 $X^{d}(t,x,i) = (X(t,x,i)-X(0,x,i))-a(x)*t$ , a field with 3 dimensions : space, time and model ensemble member. The same calculation gives you  $Y^{d}(t,x,i)$ .

If it is based on globally averaged values, I would recommend to keep the space dimension. B) With the MLR, you fitted the following relationship and thus estimated the coefficients dYd/dXd:

 $Y^{d}(t,x,i) = sum \left( (dY^{d}/dX^{d}) * X^{d}(t,x,i) \right)$

An approximation of the first order taylor development which would give

 $Y^{d}(t,x,i) = sum ( (dY^{d}/dX^{d})(t,x,i) * X^{d}(t,x,i))$ C) Then, by linearity, you compute (*t* being a 10-year average)

 $Sigma_i(Y^d(t,x,i)) = sum (dY^d/dX^d sigma_(X^d(t,x,i)))$

BUT : Variance isn't linear (neither is the standard deviation). Even if two variables are independent (which is definitely not the case),  $VAR(aX+bY) = a^2 Var(X) + b^2 Var(Y)$ .

In your case, if you wanted to reconstruct your variance, you would use the following formula (*ai* being your MLR coefficient  $dY^d/dX^d$  and Cov being the covariance and not the coefficient of variance here):

You can calculate this value perfectly well, but I'm not sure that's what you want to do.

Indeed, I'm not sure what kind of information you expect from the relationships between linearly detrended variables and linearly detrended phytoplankton carbon biomass: do you want to explain the internal phytoplankton variability by the internal variability of others variables or do you want to explain it by the trends of other variables ? I think the second option, or both, would be more appropriate (e.g., is the increase in temperature related to the reduction in internal variability of phytoplankton biomass?)

While I think your current method is wrong, I keep in mind that too few details have been provided to be certain, and perhaps I will be convinced of your method when more details are included. Nevertheless, I suggest another approach:

You have introduced CoV (Coefficient of Variance), and I think this variable is indeed more appropriate than standard deviation because it removes the effect of reduced mean state values on the change in internal variability (i.e., a lower mean state will lead to lower internal variability in absolute magnitude, but not necessarily to a reduced coefficient of variance).

$$egin{aligned} \operatorname{Var}\!\left(\sum_{i=1}^N a_i X_i
ight) &= \sum_{i,j=1}^N a_i a_j \operatorname{Cov}(X_i,X_j) \ &= \sum_{i=1}^N a_i^2 \operatorname{Var}(X_i) + \sum_{i
eq j} a_i a_j \operatorname{Cov}(X_i,X_j) \ &= \sum_{i=1}^N a_i^2 \operatorname{Var}(X_i) + 2 \sum_{1 \leq i

"Figure 4: Distribution of the dominant phytoplankton functional type in biomass carbon averaged across the RCP8.5 forcing scenario (2006 to 2100). The CESM1-LE simulates three phytoplankton functional types: diatoms, diazotrophs, and small phytoplankton. Regions where diatoms dominate are shown in yellow and regions where small phytoplankton dominate are shown in purple. Diazotrophs do not dominate in any region of the global ocean."

Review 3 of Elsworth et al. (2022) "Anthropogenic climate change drives non-stationary phytoplankton variance", submitted to Biogeosciences.

This work investigates the internal variability of phytoplankton biomass in an Earth System Model large ensemble (CESM1-LE). The results show a decrease in internal variability under RCP8.5.

The manuscript is well written and well laid out and the results are novel and important. I have only some minor suggestions.

We thank the reviewer for their constructive comments and appreciate their careful reading of the manuscript.

**Comments**

Line 128: The coefficient of variance seems to be much more commonly referred to as the coefficient of variation. Is there any reason why you use the term "coefficient of variance"?

We thank the reviewer for this clarification. We will revise the text to reflect this point.

Line 130: Should the nominator in this equation read LE(x,y,t) (with a line above) as it is defined on line 132? In that case I would rephrase the sentence on line 131 to say something like: where LE(x,y,t), the forced response of the large ensemble, is calculated as the mean of ensemble members at a given location and time.

We appreciate this suggestion. We will revise both the equation and the accompanying text to clarify this point.

Line 140: Should be biomass standard deviation instead of variance? Also on line 145. Also check other places throughout the manuscript like on lines 225-229.

Thank you for noting this discrepancy. We will revise the manuscript to make sure we are using the correct terminology at each point in the text.

Line 169 - 175: It is not specified in the text that it is surface chlorophyll that you are using. Also, could you please clarify why you used surface chlorophyll for the validation of internal variability instead of biomass as in the rest of the analysis?

Thank you for this clarification. We use surface chlorophyll for the validation of internal variability, because we are building on previous work in which we created a synthetic ensemble of observed surface chlorophyll to emulate observed variability. We will revise the text and the associated figure captions to clarify the motivation for this decision.

212: Specify that it is surface chlorophyll.

We will specify that we are referring to surface chlorophyll.

Review 4 of Elsworth et al. (2022) "Anthropogenic climate change drives non-stationary phytoplankton variance", submitted to Biogeosciences.

The authors are to be commended for taking on an interesting and important scientific challenge. However I do have a primary concern with the analytical framework applied using an MLR approach to evaluate the drivers of future changes in biomass variance. I believe that the methodology applied is sufficiently problematic that the primary result (grazing) is not convincing.

A biomass framework has been developed and applied for looking at marine ecosystems, namely that of Behrenfeld and Boss (2014; Annual Reviews), or more concisely the Behrenfeld framework. This method provides a quantitative and mechanistically-based means to connect biomass fluctuations with the underlying drivers at the timescales resolved by the model output (presumably monthly here). Within the Behrenfeld framework, decomposition into the underlying drivers is based on mass conservation using fields that should be saved for the model. For the case of the case of entrainment as it impacts biomass (a viable mechanism) there is not a convincing case that this will be properly represented by the fluctuations in annual mean MLD. This may be the case, but it is important for the authors to demonstrate this rather than simply assume it to be the case. Likewise temperature and nutrient dependencies in the model work into growth rates for phytoplankton, and the representation in the MLR approach is not convincing or state-of-the-art. The burden is really on the authors to justify their decision to use an MLR approach, rather than the mechanistically-based biomass framework of Behrenfeld.

I would urge the authors to apply the biomass framework of Behrenfeld to identify properly the relative roles of light, nutrient, and temperature drivers, with the analysis performed at the (monthly?) timescales resolved in the model output. Otherwise in my view the main ideas emphasized in the interpretation of the results are somewhat unsubstantiated.

We thank the reviewer for their constructive comments and appreciate their careful reading of the manuscript. The reviewer makes a general comment to revise our MLR analysis to make it more statistically robust. This also reflects the comments of Reviewer 1. We will address this by revising our MLR analysis to use CoV rather than standard deviation and will extend the discussion section of the manuscript to reflect his revised analysis.

While the Behrenfeld and Boss, 2014 analytical framework is applicable to the study of bloom dynamics on the scale of days to months, our analysis relates to drivers of phytoplankton variance on the scale of centuries. The difference in relevant timescales between the studies result in different analytical approaches. This study does not attempt to address drivers of bloom dynamics, but rather drivers of phytoplankton variance.

The CESM1-LE does not provide output for phytoplankton growth rate. Without information about phytoplankton growth rate, we cannot reconstruct the change in phytoplankton or the change in phytoplankton variance over time using the analytical approach suggested from Behrenfeld and Boss, 2014. However, we have explored a more statistically robust approach to reconstruct drivers of phytoplankton coefficient of variance in response to Reviewer 1's comment.

---

## Author Response (AR3)

Review 1 of Elsworth et al. (2022) "Anthropogenic climate change drives non-stationary phytoplankton variance", submitted to Biogeosciences.

This is a very interesting study on the impacts of climate change on the internal variability of phytoplankton, based on an Earth System Model ensemble. While I believe this study is quite important and of interest in terms of results (the main result being a reduction in phytoplankton internal variability under anthropogenic driven climate change) as in terms of implications(especially the link with fisheries stock assessment, whose uncertainty could be reduced as a consequence of this reduced phytoplankton internal variability), I have major concerns about the MLR method used to estimate the physical and biogeochemical drivers of trends in phytoplankton internal variability. I elaborate on these and other issues below.

**MAJOR COMMENTS:**

My first concern is the choice of explanatory variables for the MLR:

MLR is a great tool for exploring relationships between variables, but as you have indicated in the text, it is unable to distinguish between bottom-up and top-down relationships that link two variables. For these reasons, in order to be able to interpret the results with causality relationship, you should :

- 1) Use only variables for which the causal relationship with phytoplankton biomass is known (or for which the first order of this relationship is known), e.g., SST (the first order is a bottom-up relationship: warming drives phytoplankton biomass by increasing metabolic rates, a positive effect, and by increasing nutrient stratification, a negative effect. At second order, one could have a top- down feedback of phytoplankton biomass change modifying carbon cycling and indirectly temperature, but one would neglect this effect). For this reason, I think that including zooplankton/zooplankton grazing in such an analysis is not appropriate because you are not able to separate top-down and bottom-up effects on phytoplankton biomass.
- 2. 2) Use only variables for which the causal relationship with the target variable is the same. In your case, use the variables for which phytoplankton biomass is a consequence, not a cause. Again, while I believe that zooplankton do exert top-down control over phytoplankton (so that phytoplankton biomass would be a consequence), PFT models with small numbers of zooplankton are likely to be dominated by bottom-up control (so that zooplankton would be the consequence and phytoplankton the cause). To support this claim, trophic amplification under climate change in these models has been described as a good indicator of bottom-up control of zooplankton by phytoplankton (Chust et al., 2014, Kwiatkowski et al., 2019), a pattern that is altered when higher trophic levels are considered (Dupont et al., 2022).
- 3. 3) MLR analysis assumes independence of explanatory variables, which is clearly not the case (e.g., MLD and Nutrient are highly correlated). I agree that this is a classic problem in multivariate analysis on climate variables, but this point should be discussed further, by providing at least one correlation matrix between all explanatory variables.

Nevertheless, your signal on zooplankton is clearly related to the strong relationship between zooplankton and phytoplankton, which is expected but clearly interesting. I think you should analyze (with simple linear regression) the relationship between phytoplankton and zooplankton separately from the other variables, which would clearly fit the main message of your paper: showing that the effect of climate change on the internal variability of phytoplankton is transferred to the internal variability of zooplankton would demonstrate a transfer of the trend in internal variability to the higher trophic levels (in this case, zooplankton), which you could then extrapolate in the discussion to even higher trophic levels (e.g. fish). It would also be interesting to compare trends in phytoplankton and zooplankton internal variability. Is it higher? Lower? Why? You could also do the same MLR analysis with trends in total plankton with the bottom-up effect.

We thank the reviewer for their constructive comments and appreciate their careful reading of the manuscript. In response to this and another reviewer's feedback, we took a different approach to develop an understanding of the drivers of changing phytoplankton coefficient of variation (CoV) with anothropogenic climate change. We now use a machine learning approach in which we generate an ensemble of boosted regression trees to quantify drivers in changing phytoplankton carbon biomass CoV. Unlike linear models, boosted trees are able to capture non-linear interaction between the explanatory variables and the target variable. At every step, the ensemble fits a new learner to the difference between the observed response and the aggregated prediction of all learners grown previously, aiming to minimize mean-squared error.

We use the Matlab function *predictorImportance* to estimate the importance of the predictors for each tree learner in the ensemble, which computes the importance of the predicotrs in the tree by summing changes due to splits on every predictor and dividing the sum by the total number of branches. Revised estimates of the relative importance of predictor variables for changing phytoplankton CoV for each of the four key ocean regions are illustrated in a new version of Figure 5 and elaborated upon in the manuscript text.

"Figure 5: Relative importance of predictor variables on phytoplankton biomass coefficient of variance across the RCP8.5 forcing scenario (2006 to 2100). Marine ecological regions are defined in Tagliabue et al. (2021). Regions were selected which aligned with the highest fisheries catch in the (a) Atlantic and (b) Pacific basins and the biogeochemically important (c) Southern Ocean and (d) Equatorial Pacific regions. The dominant phytoplankton functional type is considered in each region. In regions with a mixed ecological assemblage, total phytoplankton carbon is considered."

L133: "We quantified the drivers of phytoplankton carbon biomass CoV in key ocean regions by generating an ensemble of boosted regression trees. Unlike linear models, boosted trees are able to capture non-linear interaction between the predictors and the response. A regression tree ensemble is a predictive model composed of a weighted combination of multiple regression trees. At every step, the ensemble fits a new learner to the difference between the observed response and the aggregated prediction of all learners grown previously, aiming to minimize mean-squared error. We generate an ensemble of boosted regression trees (maximum tree depth = 10) using the Matlab function *fitrensemble*. Our predictor variables are the regional mean, ensemble mean temperature, mixed layer depth, incoming shortwave radiation, physically mediated iron, physically mediated phosphate, zooplankton carbon, and zooplankton grazing (diatom, small phytoplankton, or their sum) annually resolved from 2006 to 2100. We use the Matlab function *predictorImportance* to estimate the importance of the predictors for each tree learner in the ensemble; it computes the importance of the predictors in a tree by summing changes due to splits on every predictor and dividing the sum by the total number of branches."

L246: "We identify the importance of different predictors to changing phytoplankton biomass CoV in four distinct ecological regions using a machine learning (boosted regression tree) approach. In the subpolar Atlantic (ASP) and subpolar Pacific (SAP) ecological provinces (Figure 4 diatom

biomass CoV declines between the beginning and end of the century (Table 1). In the Atlantic subpolar region, the most important predictor of diatom biomass CoV is phosphate advection, with smaller contributions from zooplankton carbon (Figure 5a). In the subarctic Pacific region, sea surface temperature is the most important predictor of diatom biomass CoV, with phosphate advection playing a secondary role (Figure 5b)...'

L270: "Using a machine learning approach, we identify the importance of different predictors to changing phytoplankton biomass internal variability. In all four ecological provinces, a combination of bottom-up controls (e.g., nutrient supply, light availability) and top-down controls (e.g., grazer biomass) predict the decline in phytoplankton biomass CoV with anthropogenic warming..."

**My second concern is about the MLR method itself, which I think is wrong in its current form:**

First of all, I am missing some details to understand what exactly you did with the MLR. You would need to make it clear which variables are used for each step of the method. In particular, it is not clear which variable depends on i) time, ii) space, and iii) the member of the model set. For the rest of my argument, *t* will refer to

time, x to the grid cell (spatial position) and i to the model set member. Y will refer to the phytoplankton biomass and X to any explanatory variable.

A) So, for what I understand, your first step was to prepare linearly detrended annual anomalies. So, for a variable *X*, with a trend *a*, the considered variable in the MLR is

 $X^{d}(t,x,i) = (X(t,x,i)-X(0,x,i))-a(x)*t$ , a field with 3 dimensions : space, time and model ensemble member. The same calculation gives you  $Y^{d}(t,x,i)$ .

If it is based on globally averaged values, I would recommend to keep the space dimension. B) With the MLR, you fitted the following relationship and thus estimated the coefficients dYd/dXd:

 $Y^{d}(t,x,i) = sum\left((dY^{d}/dX^{d}) * X^{d}(t,x,i)\right)$

An approximation of the first order taylor development which would give

 $Y^{d}(t,x,i) = sum ( (dY^{d}/dX^{d})(t,x,i) * X^{d}(t,x,i))$ C) Then, by linearity, you compute (*t* being a 10-year average)

 $Sigma_i(Y^d(t,x,i)) = sum (dY^d/dX^d sigma_(X^d(t,x,i)))$

BUT : Variance isn't linear (neither is the standard deviation). Even if two variables are independent (which is definitely not the case),  $VAR(aX+bY) = a^2 Var(X) + b^2 Var(Y)$ .

In your case, if you wanted to reconstruct your variance, you would use the following formula ( $a_i$  being your MLR coefficient dYd/dXd and Cov being the covariance and not the coefficient of variance here):

You can calculate this value perfectly well, but I'm not sure that's what you want to do.

Indeed, I'm not sure what kind of information you expect from the relationships between linearly detrended variables and linearly detrended phytoplankton carbon biomass: do you want to explain the internal phytoplankton variability by the internal variability of others variables or do you want to explain it by the trends of other variables ? I think the second option, or both, would be more appropriate (e.g., is the increase in temperature related to the reduction in internal variability of phytoplankton biomass?)

While I think your current method is wrong, I keep in mind that too few details have been provided to be certain, and perhaps I will be convinced of your method when more details are included. Nevertheless, I suggest another approach:

You have introduced CoV (Coefficient of Variance), and I think this variable is indeed more appropriate than standard deviation because it removes the effect of reduced mean state values on the change in internal variability (i.e., a lower mean state will lead to lower internal variability in absolute magnitude, but not necessarily to a reduced coefficient of variance).

$$egin{aligned} \operatorname{Var}\!\left(\sum_{i=1}^N a_i X_i
ight) &= \sum_{i,j=1}^N a_i a_j \operatorname{Cov}(X_i,X_j) \ &= \sum_{i=1}^N a_i^2 \operatorname{Var}(X_i) + \sum_{i
eq j} a_i a_j \operatorname{Cov}(X_i,X_j) \ &= \sum_{i=1}^N a_i^2 \operatorname{Var}(X_i) + 2\sum_{1\leq i

"Figure 4: Distribution of the dominant phytoplankton functional type in biomass carbon averaged across the RCP8.5 forcing scenario (2006 to 2100). The CESM1-LE simulates three phytoplankton functional types: diatoms, diazotrophs, and small phytoplankton. Regions where diatoms dominate are shown in yellow and regions where small phytoplankton dominate are shown in purple. Diazotrophs do not dominate in any region of the global ocean. The four ecological provinces are shown: subpolar Pacific (SAP), subpolar Atlantic (ASP), Equatorial Pacific (EQP), and Southern Ocean (SOC)."

L260: The wording of the sentence suggests that zooplankton exert top-down control over phytoplankton, which is uncertain or even false.

Thank you for this clarification. We have removed this text to reflect the results of our new machine learning analysis.

L275-279: I think this result (with an appropriate MLR method) is very interesting in explaining what is driving the zooplankton variability (see main comment) and should be interpreted from a zooplankton perspective.

Absolutely. We agree that the manuscript would benefit from applying a more statistically robust approach. We revised our statistical analysis approach to use a machine learning approach in which we generate an ensemble of boosted regression trees to quantify drivers in changing phytoplankton carbon biomass CoV. We have replaced Figure 5 with the results from our machine learning analysis and have modified the associated text to reflect this new analytical framework.

L287: I do not believe that either of these references is relevant to this statement.

Thank you for this comment. We have removed this text to reflect the results of our new machine learning analysis.

Review 2 of Elsworth et al. (2022) "Anthropogenic climate change drives non-stationary phytoplankton variance", submitted to Biogeosciences.

My only remaining recommendation is to remove Diazotrophs from the color bar for Figure 4. If Diazotrophs to not dominate in any region (as specified in the authors' responses to the initial review), it seems unnecessary to include them in the figure. If the authors have some other reason for including Diazotrophs, it should at least be specified in the caption that Diazotrophs are not actually visible in the figure.

Thank you for this suggestion. We have removed diazotrophs from the color bar in Figure 4 and have revised the figure caption. We have also revised Figure 4 to include the overlayed ecological regions considered in the manuscript.

---

## Author Response (AR5)

Review 1 of "Anthropogenic climate change drives Non-stationary phytoplankton variance", submitted to Biogeosciences.

I appreciate the dedication that the authors have put into revising the manuscript. From my perspective, with the exception of the new ML analysis, which requires further details and clarifications, the manuscript is otherwise prepared for publication pending the resolution of the two comments below. While I acknowledge the relevance of the BRT method for driver identification, I believe that additional details are necessary to instill confidence in the results. Specifically, incorporating supplementary tests could enhance the evaluation of the zooplankton's impact on phytoplankton CoV.

We thank the reviewer for their careful reading of the revised manuscript and their constructive suggestions. In response to this feedback, we have included additional detail in the methods section on the specific hyperparameters used in our machine learning approach, as well as how the model was tuned. We have also elaborated on the machine learning model's performance by revising Figure 5 to include RMSEs for each of the four regional analyses.

Additionally, the reviewer makes a general comment regarding the effect of zooplankton grazing controls on the predictive skill of the model. To address this comment, we withheld all zooplankton grazing terms (zooplankton carbon, diatom grazing, and small phytoplankton grazing) when performing the predictor importance analysis in the Equatorial Pacific (the only region with a strong zooplankton dependence). When zooplankton grazing terms were withheld in this region, the RMSE increased by 7%, indicating that the predictive model performs slightly worse without zooplankton grazing included.

Comment #1: ML Methods

Though I lack expertise in ML, it seems that insufficient information is provided to comprehensively interpret the new findings. Recognizing that the paper's main focus might not be on this aspect, it's essential to provide a rational explanation for the utilization of this method rather than treating it as a "black box."

Specifically:

- Could you elaborate on the model's performance? Visualizing a time series would help confirm whether BRT effectively reconstructs CoV time series. Metrics like RMSE and r2 for both training and testing sets would also provide clarity.

Thank you for this suggestion. We have elaborated on the model's performance by including the RMSE for the testing dataset in Figure 5 for each region analyzed using the machine learning method. We have noted the addition of the RMSE in the figure caption.

[Figure]

"Figure 5: Relative importance of predictor variables on phytoplankton biomass coefficient of variation across the RCP8.5 forcing scenario (2006 to 2100). Marine ecological regions are defined in Tagliabue et al. (2021). Regions were selected which aligned with the highest fisheries catch in the (a) Atlantic and (b) Pacific basins and the biogeochemically important (c) Southern Ocean and (d) Equatorial Pacific regions. The dominant phytoplankton functional type is considered in each region. In regions with a mixed ecological assemblage, total phytoplankton carbon is considered. The RMSE (mmol C m$^{-2}$) for the testing dataset of each machine learning analysis is included in the upper right corner of each panel."

- Additionally, showing partial dependency plots (examples in Dannouf et al. (2022) or Lamb et al. (2021)), could help elucidate each variable's contribution.

Thank you for bringing this to our attention. Partial dependency plots are key to understanding the contribution of each predictor variable when using other 'black box' statistical/machine learning approaches (e.g. Gaussian Process Regression Models or Neural Networks). However, since we apply a Boosted Regression Tree which implicitly provides predictor importance, we can effectively reconstruct the relative importance of all predictor variables on phytoplankton carbon without the use of a partial dependency analysis. However, we now include the RMSE of our testing dataset from our machine learning approach in Figure 5 to elaborate on the model's performance.

- You would also need to specify how you tuned the model (i.e. how you choose the hyperparameters: learning rate, depth, number of trees).

We agree that the manuscript would benefit from more detail on how the machine learning model was tuned. We have added text to the methods to describe how the machine model was tuned and to include the specific hyperparameter values used.

L32: "The machine learning model was tuned to a learning rate of 1 and a tree depth of 10, generating 100 trees. We tuned several hyperparameters to generate the highest quality predictive results with the least computational expense. While learning rate can affect the quality of the solution, we experimented with a range of learning rates (0.1-1) with no change in the predictive results. Similarly, we tuned the tree depth using a range of 1 to 10 splits, and tree depths less than 10 produced a higher RMSE of the testing dataset."

- Clarification is needed on whether your predictors are regional time series spanning 2006 to 2100, as hinted at in L138-141.

Thank you for this feedback. We have modified the text to clarify the temporal extent of our regional analyses.

L26: " Our predictor variables are the regional mean, ensemble mean temperature, mixed layer depth, incoming shortwave radiation, physically mediated iron, physically mediated phosphate, zooplankton carbon, and zooplankton grazing (diatom, small phytoplankton, or their sum) annually resolved from 2006 to 2100, while our response variable is CoV of phytoplankton carbon (diatom, small phytoplankton, or their sum) annually resolved from 2006 to 2100."

I believe some refinement of the references is necessary to better justify the application of this method. There are a few reference suggestions that could help in establishing a more precise methodology (although there could be more relevant references). I think that integrating the BRT analysis into this paper could be better supported without significantly increasing its length, by fairly utilizing the supplemental information as a means of support. Have a look at Elith et al. (2008) for an introduction on the ecological application of BRT. For insights into BRT applied to time series, Dannouf et al. (2022) and Lamb et al. (2021) could be valuable references. While Denvil-Sommer (2023) focuses on the application of ML to ESM simulated spatial data (rather than temporal), there's potential inspiration for method structure. Additionally, consider referring to Robert et al. (2017) for insights into cross-validation.

Elith, J., Leathwick, J.R., & Hastie, T. (2008). A working guide to boosted regression trees. Journal of animal ecology, 77(4), 802-813.

Lamb, S.E., Haacker, E.M.K., & Smidt, S.J. (2021). Influence of irrigation drivers using boosted regression trees: Kansas High Plains. Water Resources Research, 57, e2020WR028867. https://doi.org/10.1029/2020WR028867

Denvil-Sommer, A., Buitenhuis, E.T., Kiko, R., Lombard, F., Guidi, L., & Le Quéré, C. (2023). Testing the reconstruction of modelled particulate organic carbon from surface ecosystem components using PlankTOM12 and machine learning. Geoscientific Model Development, 16(10), 2995-3012.

Dannouf, R., Yong, B., Ndehedehe, C.E., Correa, F.M., & Ferrerira, V. (2022). Boosted Regression Tree Algorithm for the Reconstruction of GRACE-Based Terrestrial Water Storage Anomalies in the Yangtze River Basin. Frontiers in Environmental Science, 10, 917545.

Roberts, D.R., Bahn, V., Ciuti, S., Boyce, M.S., Elith, J., Guillera-Arroita, G., … & Dormann, C.F. (2017). Cross-validation strategies for data with temporal, spatial, hierarchical, or phylogenetic structure. Ecography, 40(8), 913-929.

Thank you for providing these very helpful resources. We have included them in the text to highlight machine learning methodologies in more detail.

L21: "Unlike linear models, boosted trees are able to capture non-linear interaction between the predictors and the response, and have been used in a number of ecological applications (Elith et al., 2008; Roberts et al., 2016; Lamb et al., 2021; Dannouf et al., 2022; Denvil-Sommer et al., 2023)."

Comment #2: Predictor choice (in particular zooplankton)

However, even using BRT, it's important to clarify that having zooplankton as a predictor doesn't necessarily mean it's the cause. The only way to really test the nature of the relation would be to run the model without zooplankton, which I know would require substantial extra work, so that I don't think it is necessary. Despite this, my reservations regarding the utilization of zooplankton grazing and zooplankton biomass predictors persist. If it's more comfortable to designate light and nutrients as "bottom-up control," it might be less accurate to term grazing and zooplankton biomass as "top-down control" (as mentioned in L10, L259, L272). This is because their changes during the focus period could also reflect variations in phytoplankton. Here are some suggestions that could assist in identifying the role of zooplankton as a top-down driver of phyto CoV in different regions and reinforcing your assumptions:

We thank the reviewer for this comment. To test the effect of the zooplankton grazing controls, we withheld all zooplankton terms (zooplankton carbon, diatom grazing, and small phytoplankton grazing) when performing the predictor importance analysis in the Equatorial Pacific (the only region with a strong zooplankton dependence). When zooplankton grazing terms were withheld in this region, the RMSE increased by 7%, indicating that the predictive model performs slightly worse without zooplankton grazing included. We have opted to maintain "top-down" and "bottom-up" to describe the controls on phytoplankton biomass, as these phrases are regularly used in the literature. This terminology is commonly used in both the ocean biogeochemistry literature (e.g., Bopp et al., 2001; Hashioka et al., 2013; Prowe et al., 2011; Behrenfeld et al., 2010, 2013; Laufkötter et al., 2015) and in its seminal textbook (Sarmiento and Gruber, 2006).

- What does the correlation matrix between predictors look like, particularly for phytoplankton, zooplankton, and grazing?

As mentioned above, we withheld the zooplankton terms and recalculated the RMSE of the testing dataset when developing a BRT model in the Equatorial Pacific, where zooplankton plays a key

role. When zooplankton grazing terms were withheld in this region, the RMSE increased by 7%, indicating that the predictive model performs slightly worse without zooplankton grazing included.

- When you apply BRT using the same set of predictors but replace zooplankton biomass with phytoplankton biomass or Chl a, do you obtain similar results (i-e does the importance of phytoplankton matches that of zooplankton as a predictor)? If so, it would suggest that top-down control is unlikely.

As mentioned above, we withheld the zooplankton terms and recalculated the RMSE of the testing dataset when developing a BRT model in the Equatorial Pacific, where zooplankton plays a key role. When zooplankton grazing terms were withheld in this region, the RMSE increased by 7%, indicating that the predictive model performs slightly worse without zooplankton grazing included.

- How are you defining "grazing pressure"? Is it the total amount of grazed phytoplankton, or is it normalized by phytoplankton biomass? I believe the second option might be more suitable to account for zooplankton's top-down influence.

In this context, grazing pressure is the fraction of phytoplankton biomass grazed. We have defined grazing pressure in the discussion to clarify this point. We also point the reviewer to the methods and supplemental information where we have discussed the functional form of zooplankton grazing in the CESM1-LE.

L30: "Previous studies of phytoplankton change with climatic warming have demonstrated that grazing pressure, the fraction of phytoplankton biomass grazed, is a contributor to biomass decline in low to intermediate latitude regions across a suite of model simulations with different marine ecosystem models (Laufkötter et al., 2015)…"

- How does the performance of the ML model improve when you include zooplankton/grazing compared to an ML model with only bottom-up controls.

Thank you for this comment. To test the effect of the zooplankton grazing controls, we withheld all zooplankton terms (zooplankton carbon, diatom grazing, and small phytoplankton grazing) when performing the predictor importance analysis in the Equatorial Pacific (the only region with a strong zooplankton dependence). When zooplankton grazing terms were withheld in this region, the RMSE increased by 7%, indicating that the predictive model performs slightly worse without zooplankton grazing included.

- Does the BRT's performance show enhancement when you train the model regionally compared to using a global scale?

Thank you for this suggestion. We chose not to train the machine learning model on the global scale as regionally specific processes dominate in each ecosystem. However, our regional analyses allow us to identify predictive drivers in discrete regional ecosystems with cohesive ecological and biogeochemical dynamics.

- In a broader context, similar to Denvil-Sommer et al. 2023, you could experiment with different sets of predictors to observe how the model performs and gain insights into the most crucial drivers. Given the critical nature of predictor choice in ML, this could be particularly informative for testing the role of zooplankton.

As mentioned above, we withheld the zooplankton terms and recalculated the RMSE of the testing dataset when developing a BRT model in the Equatorial Pacific, where zooplankton plays a key role. When zooplankton grazing terms were withheld in this region, the RMSE increased by 7%, indicating that the predictive model performs slightly worse without zooplankton grazing included.